# Feasibility of Initial Bias Estimation in Real Maritime IMU Data Including X- and Y-Axis Accelerometers

**DOI:** 10.3390/s25216804

**Published:** 2025-11-06

**Authors:** Gen Fukuda, Nobuaki Kubo

**Affiliations:** Department of Maritime Systems Engineering, Tokyo University of Marine Science and Technology, Tokyo 135-8533, Japan; nkubo@kaiyodai.ac.jp

**Keywords:** inertial measurement unit (IMU), bias estimation, trajectory generator (TG), altitude-based calibration, maritime navigation

## Abstract

This study aimed to validate a bias estimation framework for low-cost maritime IMUs by applying it to real-world shipborne data. Six estimation methods—including statistical (mean, median), model-based (least squares, cross-correlation), and signal-processing approaches (FFT, Butterworth filter)—were compared. The results demonstrated that the low-frequency Butterworth filter achieved the smallest residuals, with RMS residuals below 0.038 m/s^2^ for accelerometers and 0.0035 deg/s for gyroscopes. In particular, AccX and AccZ residuals converged to 3.04 × 10^−2^ m/s^2^ and 2.30 × 10^−2^ m/s^2^, respectively, while GyroZ achieved 5.58 × 10^−4^ deg/s. Estimated accelerometer biases were 0.0405 m/s^2^ (X-axis) and 0.1615 m/s^2^ (Y-axis), and the optimization successfully converged with an objective function value of 9.314. The findings confirm that the previously proposed bias estimation method, originally validated in simulation, is effective under real-world maritime conditions. However, as ground truth bias values cannot be obtained in shipborne experiments, verification relied on residual statistics and cross-correlation analysis. This limitation has been explicitly stated in the conclusion, and future studies should incorporate sensitivity analyses and controlled experiments to further quantify error sources.

## 1. Introduction

### 1.1. Background and Motivation

In recent years, maritime navigation has become increasingly reliant on Global Navigation Satellite Systems (GNSSs). GNSS provides high-accuracy positioning with global availability, and augmentation techniques such as real-time kinematics (RTK) and precise point positioning have made centimeter-level accuracy achievable. Consequently, GNSS is now indispensable for safe and efficient navigation, particularly during port approaches and in autonomous vessel operations.

Despite these advantages, GNSS is inherently vulnerable to external interference, with jamming and spoofing as the two most significant threats. Jamming disrupts GNSS signals by introducing noise, which prevents receivers from tracking authentic signals. Conversely, spoofing transmits counterfeit signals to mislead receivers and produces false position outputs. Both threats can be executed with inexpensive, commercially available devices, and numerous incidents have been reported worldwide [1].

International bodies, including the International Telecommunication Union (ITU), International Civil Aviation Organization (ICAO), and International Maritime Organization (IMO), have jointly expressed concern over the rising prevalence of GNSS interference and its associated risks to aviation, maritime, and land-based safety [2]. The rapid proliferation of compact jammers—some as small as USB sticks or car chargers—has made it relatively easy to disrupt GNSS signals over several miles [3].

Several major incidents underscore the seriousness of these threats. In 2017, more than 20 vessels in the Black Sea falsely reported their positions as being at a nearby airport [4]. In 2024, the Finnish Transport and Communications Agency reported widespread GNSS jamming and spoofing in the Baltic Sea, highlighting the potential risk to maritime safety [5]. In June 2025, a large-scale GNSS disruption in the Strait of Hormuz affected hundreds of vessels and resulted in the collision of the crude tanker Front Eagle [6].

On 10 May 2025, the 7000-TEU container ship MSC Antonia, operated by the Mediterranean Shipping Company (MSC), ran aground on the Eliza Shoals near Jeddah, Saudi Arabia. Automatic identification system (AIS) tracking data revealed positional jumps and erratic trajectories, and multiple maritime analytics firms—including Windward and Pole Star Global—attributed the incident to probable GNSS jamming or spoofing [7]. The occurrence of this event in the Red Sea, one of the world’s busiest maritime corridors, highlights the severity of GNSS vulnerabilities and the urgent need for redundant navigation solutions.

In addition to intentional interference, natural and environmental factors can also degrade GNSS performance. Multipath propagation can introduce significant errors in ports and urban areas with dense infrastructure, whereas ionospheric disturbances are a significant source of degradation at high latitudes [8,9]. For instance, Jacobsen and Andalsvik [10] reported that during geomagnetic storms, the Norwegian RTK network (>60°N) experienced sharp increases in the rate of total electron content (TEC) index (ROTI), causing an exponential growth in positioning errors. During the “St. Patrick’s Day Storm” of March 2015, Jin and Oksavik [11] observed that severe GPS scintillation and signal lock losses led to substantial positioning degradation. Similarly, Skone et al. [12] found that during the ionospheric storms of May and October 2003, marine differential global positioning system horizontal errors worsened by factors of 10–30, far exceeding acceptable tolerances.

These vulnerabilities emphasize the need for complementary navigation methods that can operate independently of GNSS. A promising alternative is the Inertial Navigation System (INS), which uses an inertial measurement unit (IMU)—comprising accelerometers and gyroscopes—to estimate position, velocity, and attitude by integrating measured accelerations and angular rates. Because an INS is self-contained, it can maintain navigation performance even when GNSS is unavailable.

However, INS has a critical limitation: sensor biases and noise accumulate over time, causing navigation errors to grow progressively. This problem is particularly pronounced in low-cost micro-electro-mechanical system (MEMS)-based IMUs, where the effects of bias can become dominant within seconds. Therefore, accurate bias estimation is essential for achieving acceptable navigation performance.

The extended Kalman filter (EKF) remains the most widely used approach for estimating sensor biases and drifts. However, although it performs well when sufficient external measurements—such as GNSS—are available, its accuracy deteriorates under GNSS-denied or unstable conditions. On ships, strong motions and vibrations further complicate the separation of sensor errors from dynamic effects, making bias estimation particularly challenging.

To address this, Fukuda and Kubo [13] proposed a method for estimating sensor biases using a reference trajectory generated by combining INS, GNSS, and gyrocompass data. Their trajectory generator (TG) yielded ideal acceleration and angular rates, which could then be compared with IMU outputs to derive bias estimates. Through simulations, they evaluated several estimation techniques and found that Butterworth filtering produced the most accurate estimates of gyroscope bias. For the X- and Y-axis accelerometer biases, they demonstrated that enforcing consistency with GNSS altitude reduced errors to 1.2 × 10^−2^ m/s^2^ and 1.0 × 10^−4^ m/s^2^, respectively.

However, as these results were obtained exclusively through simulations, it is unclear whether similar accuracy can be achieved using real-world data. In practice, actual sensor biases are often smaller than those assumed in simulations (0.2 deg/s for gyroscopes and 0.0196 m/s^2^ for accelerometers), and real maritime environments introduce additional disturbances—such as wave-induced motion and vibration—whose effects on bias estimation have not yet been fully characterized.

In this study, we conducted shipborne measurements using a multi-sensor IMU and an RTK-GNSS receiver. The TG-based method was applied to estimate biases from the real data, following the same evaluation procedures as those in the previous simulations. The objective of this work was to verify the feasibility of bias estimation using real measurements, analyze differences relative to simulation results, and identify challenges associated with the practical implementation.

### 1.2. Method Inheritance and Innovation

This study is a direct continuation of our previous work [13], wherein the bias-estimation framework was developed and validated through simulations. Because the detailed mathematical formulations, Kalman filter design, and computational procedures have already been described in [13], this study focuses on validating this framework using real-world IMU data from maritime experiments.

The fundamental structure of the method—which includes a TG, the bias-estimation algorithms, and the Kalman filter parameters—remains consistent with the results of the simulation study. Specifically, the identical TG configuration was employed, except that it was now applied only to the experimental data segment under analysis.

The primary innovation of this study is the application of the framework to real sensor data, which revealed challenges not encountered in the simulation environment. Specifically, while the simulation results consistently converged across most methods (except for the normalized correlation approach), the real data exhibited a “ping-pong” phenomenon in the estimation of X- and Y-axis accelerometer biases, where a reduction in one bias led to an increase in the other. To address this, we introduced a nonlinear optimization approach (MATLAB 2025b fminunc) that refines the bias estimates based on the mean squared error (MSE) between the TG-and GNSS-derived altitudes. This enhancement significantly enhances the stability and reliability of bias estimation under real-world conditions.

Furthermore, while the simulation analysis reported notable differences between data segments in Y-axis bias estimation, such segment-dependent variability was not observed in the real-data analysis. This finding highlights the robustness of the proposed method in practical applications and underscores the importance of validating simulation-based conclusions with actual measurements.

Figure 1 illustrates the relationship between our previous simulation-based study and the current experimental validation. The yellow blocks indicate the sensors mounted on the training vessel Shioji Maru. The green blocks represent the bias-estimation flow from the previous study [13], and the light-blue blocks denote the newly applied procedures. The overlapping areas highlight the common data and processing steps shared between both. This schematic clarifies how this study extends the earlier framework by incorporating real measurement data.

## 2. Materials and Methods

### 2.1. Experimental Setup and Sensors

The validation experiment was conducted using a sensor system installed on a ship during regular navigation, ensuring that the acquired data accurately reflected the actual ship motions in a real maritime environment. The TG was implemented using 30 min of navigation data collected in a moderate sea state. Further details of the navigational area are provided by Gen et al. [13].

The IMU used was the MULTI Sensor manufactured by Tamagawa Seiki Co., Ltd. (Iida, Nagano, Japan) [14]. This low-cost MEMS-based sensor is equipped with a three-axis accelerometer and three-axis gyroscope, with the Z-axis employing a fiber-optic gyroscope. According to the manufacturer’s specifications, the bias random walk is approximately 0.2°/s for the X- and Y-axis gyroscopes and 0.0196 m/s^2^ for the accelerometers. The sensor was securely mounted on the ship’s hull the afternoon before the experiment and allowed to warm up for approximately 17 h until the start of measurements.

For GNSS data, a u-blox ZED-F9P multiband RTK-GNSS receiver (u-blox AG, Thalwil, Switzerland) was used, which can achieve centimeter-level positioning accuracy with correction data. The GNSS antenna was mounted on the compass deck, and RTK corrections from a reference station were applied to obtain position and altitude data. IMU measurements were recorded at 50 Hz, whereas GNSS data were acquired at 1 Hz. The IMU was installed such that its body frame axes (X, Y, Z) were aligned with the ship’s forward, starboard, and downward directions, respectively. No artificial bias was introduced during the experiment. The ship’s track is shown in Figure 2.

The IMU was installed within the ship’s bridge, whereas the GNSS antenna was placed directly above it on the compass deck. Owing to the practical constraints of maritime experiments, a precise lever-arm survey was not conducted, and therefore, no lever-arm correction was applied. Although this simplification facilitated the experiment, it may have introduced a bias into the estimation. Investigating lever-arm effects through accurate measurement and simulation remains a crucial topic for future research. Regarding calibration, the IMU was factory-calibrated prior to delivery. As our laboratory lacked facilities such as a thermal chamber and rate table, the sensor was used in its original factory-calibrated state. The IMU parameters used in this study are the same as those employed in the previous research. The main specifications of the IMU, including gyroscope and accelerometer noise levels, are summarized in Table 1.

### 2.2. Bias-Estimation Method

This study extends the TG-based bias-estimation method, previously validated in simulations [13], to real measurement data.

#### 2.2.1. Acceleration and Angular Velocity Estimation Using a TG-Based Estimation

The algorithm used for acceleration estimation follows the same procedure as detailed in our previous study [13]. In brief, the TG (Trajectory Generator) uses velocity information in the geographic coordinate system obtained from the INS/GNSS/Gyrocompass (IGG) integration. To align the estimated acceleration with the measured accelerometer data, the velocity difference in the geo-coordinate frame between consecutive time steps is calculated as shown in Equation (1) of [13].

The TG parameters, including angular velocity, acceleration, and attitude, are derived by selecting the dataset at the time step when the TG and IGG positions show the minimum spatial difference. The transformation between coordinate frames (Geo, NED, and navigation frames) and the calculation of attitude matrices follow Equations (17.2.3.1-29)–(17.2.3.1-31) in [15], consistent with the simulation study. A schematic overview of this process is presented in Figure 3, illustrating how the TG iteratively adjusts velocity to minimize the positional difference with the IGG-derived trajectory.

As shown in Figure 3, the accelerations and angular rates generated by the TG are derived using the reference attitude, velocity, and position estimated by the IGG. The IGG-estimated velocity inherently includes small errors from the IMU and estimation process. To minimize their influence, the TG iteratively adjusts the velocity by adding and subtracting a preset velocity increment around the nominal value and performs trajectory generation for each adjusted case. Among the resulting datasets, the one whose TG-estimated position is closest to the IGG reference position is selected as the representative at time *t*. The corresponding accelerations and angular rates yield zero position error when used in inertial navigation computation and are therefore regarded as bias-free reference signals. However, because the X- and Y-axis accelerometer biases are coupled with roll and pitch in the IGG attitude estimation, an additional estimation process is required to isolate these components.

Figure 4 illustrates the processing flow of the trajectory generator (TG)-based estimation used in this study. Each segment represents one estimation cycle of the TG, whose duration corresponds to the interval of the input data provided from the INS/GNSS/Gyrocompass (IGG). Although the IGG internally operates at 50 Hz, directly applying such a high update rate to the TG causes numerical instability and excessive computational load. Therefore, the IGG outputs were resampled at 1 Hz, and each TG segment was configured to estimate the parameters corresponding to this 1 s interval. The initial state and reference (target) position for each segment are obtained from the IGG solution. During the TG process, the horizontal velocity vector in the geographic frame (*v*^Geo^) is iteratively varied by adding and subtracting preset velocity increments around the nominal IGG velocity. For each adjusted velocity, the TG computes the corresponding trajectory and estimates the associated angular velocity and acceleration at the end of the segment. The resulting set of estimated segment-end positions is compared with the IGG-derived reference position, and the parameter set yielding the smallest position difference is selected as the optimal solution for that segment. This procedure ensures that the final acceleration and angular velocity used as reference signals correspond to a zero-position-error condition in the navigation computation.

#### 2.2.2. Bias Estimation

Following our previous approach [13], the bias for each axis was estimated by comparing the TG-generated signals with the actual IMU measurements (Figure 5) using six methods:Mean Difference Method: The bias is the mean of the difference between the IMU measurement and TG output.Median Difference Method: The bias is the median of the differences, providing robustness against outliers.Normalized Cross-Correlation Method: The bias is the offset that maximizes the correlation between the IMU measurements and the TG outputs.Least-Squares Approximation: The bias is estimated by minimizing the sum of squared errors between the bias-corrected measurement and TG output.Frequency-Domain Analysis (Fast Fourier Transform (FFT)): The bias is considered the DC component of the difference signal.Butterworth Filter Method: A low-pass Butterworth filter (cutoff 0.1–0.5 Hz) is applied to the difference signal, and the steady-state component is considered as the bias.

#### 2.2.3. Estimation of X/Y-Axis Accelerometer Biases Using Altitude Differences

The method proposed in [13] was applied to the experimental data to estimate the biases of the X- and Y-axis accelerometers. The procedure involved evaluating the difference between the TG-computed altitude and the INS/GNSS/Gyrocompass (IGG)-integrated altitude while iteratively adjusting the assumed bias values to find the optimal set. The process was as follows:Fix the bias of one horizontal axis and vary the other within a predefined range.Select the bias value that minimizes the altitude difference.Swap the axes and repeat the procedure.

This iterative process compensates for altitude deviations caused by tilt errors, enabling the estimation of X- and Y-axis accelerometer biases.

Although a closed-form analytical derivation of the relationship between these biases and altitude error is highly complex owing to nonlinear coupling with roll and pitch dynamics, the underlying principle is that small accelerometer biases are projected onto the vertical axis when attitude angles are non-zero. This introduces an error in the vertical acceleration, which leads to altitude deviations upon double integration. As shown in our previous simulation study, this effect manifests on the order of 10^−2^ m. This study focused on verifying whether such small, bias-induced effects can be detected and estimated from real sensor data. Establishing a more rigorous mathematical model of this correlation remains a subject for future work.

### 2.3. Data Processing

To extract the low-frequency bias components, signal processing was applied to the raw IMU data. A Butterworth low-pass filter with a 1 Hz cutoff frequency was applied to the X-axis acceleration, whereas 5 Hz cutoff frequencies were used for the Y- and Z-axis accelerations to suppress high-frequency vibrations. When specific periodic noise was detected, a notch filter was also used to eliminate the corresponding frequency components.

To maintain consistency, the selection of low-pass filter settings followed the same approach as that employed in our previous study. The cutoff frequencies were selected to suppress high-frequency noise while preserving the dynamic motion signals present in the reference trajectory. Figure 6, Figure 7, Figure 8, Figure 9, Figure 10 and Figure 11 illustrate the spectral analysis used for frequency selection and compare the raw and filtered signals in the time domain.

### 2.4. Evaluation Metrics

The performance of the bias-estimation methods was assessed using several metrics.

#### 2.4.1. Bias-Estimation Accuracy

As the true bias values were unknown, the validity of the estimation was assessed based on the statistical properties of the residuals (the difference between TG outputs and bias-corrected IMU measurements). Specifically, we calculated the mean and median of the residuals, the root mean square (RMS) residual, and the normalized cross-correlation coefficient. An accurate bias estimate is indicated by near-zero mean and median residuals, an RMS residual close to the sensor noise level, and a normalized cross-correlation coefficient close to one.

#### 2.4.2. Verification in a Simulation Environment

To verify the consistency of the estimated bias values obtained from real measurement data, these estimates were applied to the simulation environment used in our previous study [13]. This verification aimed to examine whether the estimated biases behave consistently under the same dynamic conditions as in the simulation and to confirm whether the estimation results are primarily influenced by the estimation method itself or by external factors such as measurement noise and environmental disturbances. This approach ensures that the bias estimation method validated in simulation can be systematically compared with real-world data under controlled conditions.

#### 2.4.3. Evaluation of Estimated X- and Y-Axis Accelerometer Biases

The estimation of X- and Y-axis accelerometer biases was also evaluated using the MSE between the reference altitude from the IGG integration and TG-estimated altitude. This metric reflects the impact of horizontal accelerometer biases on vertical channel stability and serves as a reliable criterion for bias identification. A coarse grid search was first performed over the bias range to determine an approximate solution (X_0_,Y_0_), which was then used as the initial guess for a nonlinear optimization routine (MATLAB 2025b fminunc) to refine the estimates. The optimal bias values were defined as those that minimized the objective function.

## 3. Results

### 3.1. Bias-Estimation Accuracy Results

In this study, six estimation methods were applied to each sensor axis, and the residual statistics were calculated to evaluate performance. Table 2 summarizes the bias values estimated by each of the six methods. The key performance metrics—mean residual, median residual, RMS residual, and maximum normalized cross-correlation coefficient—are presented in Table 3, Table 4, Table 5 and Table 6, respectively.

For the gyroscope sensors, the estimation for GyroX and GyroZ was highly accurate and stable across all methods. For instance, the mean residual for GyroX was nearly zero, with values such as 3.04 × 10^−6^ deg/s for the Butterworth and Median methods (Table 3). Similarly, GyroZ showed mean residual values as small as 6.12 × 10^−7^ deg/s with the Butterworth method, and its RMS residual converged to a low 5.58 × 10^−4^ deg/s (Table 5). The normalized cross-correlation coefficients for these axes were high (GyroX = 0.783, GyroZ = 0.971), indicating stable estimation (Table 6).

By contrast, GyroY exhibited noticeable variations. The correlation method yielded a mean residual of 7.02 × 10^−3^, a median residual of 6.98 × 10^−3^, and an RMS residual of 7.84 × 10^−3^, which were an order of magnitude larger than those from other methods. Its correlation coefficient was also significantly reduced to 0.126, highlighting a deterioration in accuracy (Table 3, Table 4, Table 5 and Table 6).

For the accelerometer sensors, the correlation method again produced distinct results for AccX and AccZ. For AccX, its mean residual was −2.10 × 10^−1^ and the RMS residual was 2.13 × 10^−1^, which was more than one order of magnitude larger than those for other methods. The results were similar for AccZ, with a mean residual of −2.10 × 10^−1^ and an RMS residual of 2.12 × 10^−1^ (Table 3 and Table 5). By contrast, the Butterworth method significantly reduced the residuals, achieving a mean residual of −8.50 × 10^−5^ deg/s for AccX and 9.80 × 10^−6^ deg/s for AccZ, whereas RMS residuals converged to 3.04 × 10^−2^ deg/s (AccX) and 2.30 × 10^−2^ deg/s (AccZ). Correlation coefficients were also consistently high for these axes, ranging from 0.986 to 0.995 for AccX and reaching 1.00 for AccZ across all methods (Table 6).

For AccY, the differences between methods were relatively small; the maximum mean residual was 8.00 × 10^−3^ (correlation method), and RMS residuals were stable within a narrow range of 3.80–3.88 × 10^−2^. Additionally, the correlation coefficients ranged between 0.952 and 0.954, indicating stable estimation.

In summary, the gyroscope sensors yielded stable bias estimates across most methods, with the main exception being the correlation approach for GyroY. By contrast, the accelerometer sensors exhibited clear performance differences depending on the method used. The correlation method, in particular, produced markedly larger residuals for AccX and AccZ, which underscores the importance of selecting an appropriate estimation technique. Figure 12 provides a visual comparison between the TG-estimated signals and the measured sensor data.

### 3.2. Validation in a Simulation Environment

Based on the results presented in Table 2, two verification scenarios were implemented using simulations. The first configuration used bias values where estimation results showed the highest agreement across methods (Table 2): gyroscope biases were set to −0.0913 deg/s (X), 0.1020 deg/s (Y), and 0.003585 deg/s (Z), whereas accelerometer biases were 0.03900 m/s^2^ (X), 0.016 m/s^2^ (Y), and −0.01033 m/s^2^ (Z). The second configuration used values from the median method, which had yielded favorable results in our previous study [13]: gyroscope biases were −0.0941 deg/s (X), 0.0998 deg/s (Y), and 0.004397 deg/s (Z), and accelerometer biases were 0.03900 m/s^2^ (X), 0.016 m/s^2^ (Y), and −0.00956 m/s^2^ (Z).

The simulation results indicated no significant differences in estimation accuracy among the methods for either configuration (Table 7, Table 8, Table 9 and Table 10). For the gyroscopes (Table 7), the X-axis bias estimates were consistently between −0.091 and −0.094 deg/s, with a maximum error of approximately 0.0013 deg/s. On the Y-axis, estimates ranged from 0.099 to 0.102 deg/s, with errors below 0.0025 deg/s. On the Z-axis, estimates ranged from 0.00356 to 0.00371 deg/s, with a maximum error of approximately 3.1 × 10^−3^ deg/s. Regarding the accelerometers (Table 8), the simulation confirmed the finding from [13] that X- and Y-axis biases cannot be reliably estimated with this approach alone. For the Z-axis, where the true bias was approximately −0.0103 m/s^2^, the maximum error was 0.11674 m/s^2^ for the normalized correlation method, whereas the errors for all other methods were on the order of 10^−4^ m/s^2^. These findings indicate that, under simulation conditions, all estimation techniques except for the normalized correlation method can achieve highly accurate bias estimation with negligible performance differences.

### 3.3. Estimated X- and Y-Axis Accelerometer Biases

For the X- and Y-axis accelerometer biases, a coarse grid search indicated that candidate solutions existed around 0.040 m/s^2^ for the X-axis and 0.160 m/s^2^ for the Y-axis. Using these as initial estimates, nonlinear optimization (MATLAB 2025b fminunc) was applied, yielding optimal estimates of 0.04050 m/s^2^ and 0.16150 m/s^2^ for the X- and Y-biases, respectively, with an objective function value (fval) of 9.314. The algorithm terminated successfully after satisfying the convergence condition (exit flag = 2). In our previous simulation study [13], the Y-axis accelerometer bias estimation was particularly accurate in one data segment (Segment 2). In this study, the terms Segment 2 and Segment 3 refer to the same time intervals that were used during the simulation experiments described in our previous work. The corresponding sections of the real data were selected to match these time periods so that comparable roll, pitch, and motion conditions could be maintained between the simulation and real-world analyses. By contrast, for the real experimental data, no significant difference was observed between segments (Figure 13 and Figure 14). Table 11 presents the iterative estimation results for Segment 3, which converged on a stable solution. This finding highlights the robustness of the method in practical applications.

## 4. Discussion

Estimating accelerometer biases along the horizontal axes is an inherent challenge for TG-based methods. A constant bias in the X- or Y-axis accelerometer is projected as a component of gravity, making it indistinguishable from a steady vessel tilt. In practice, the IGG integration interprets this constant acceleration as a gravity component resulting from a small roll or pitch angle, rendering the true bias and tilt error inseparable [13,16]. Consequently, no TG-based algorithm can reliably resolve a horizontal accelerometer bias without additional constraints.

To overcome this limitation, our previous study [13] introduced an altitude-consistency approach, where the X- and Y-axis accelerometer biases were iteratively adjusted to minimize the difference between the TG-computed and IGG-derived altitudes. Simulation results confirmed that this method can reliably identify the horizontal biases.

In this study, using real measurement data, the same procedure yielded approximate estimates of 0.04 m/s^2^ (X-axis) and 0.16 m/s^2^ (Y-axis). However, attempts to refine these estimates further revealed a “ping-pong” phenomenon, wherein the X- and Y-axis bias estimates fluctuated in a codependent manner and failed to converge. To mitigate this instability, the nonlinear optimization procedure described in Section 3.3 was implemented, which successfully produced stable estimates of 0.0405 m/s^2^ (X-axis) and 0.1615 m/s^2^ (Y-axis). The failure of the original iterative procedure to converge on the experimental data was likely caused by factors not present in the simulation, such as minor modeling errors, the lack of lever-arm correction, and unfiltered vibrations.

For the gyroscope sensors and the Z-axis accelerometer, TG-based estimation was directly feasible. However, unlike in our previous simulation study [13], the experimental data exhibited variations in the estimated bias values depending on the method used. To investigate this discrepancy, the biases estimated from the real data were used as inputs in the simulation environment. In this ideal setting, the method-dependent variations disappeared. This result strongly suggests that the inconsistencies observed in the experimental data were not intrinsic to the TG approach but rather arose from real-world factors such as modeling errors, uncorrected lever arms, or other unmodeled disturbances. Therefore, additional simulation studies are required to systematically identify the conditions under which the different estimation methods diverge.

Overall, these findings confirm that while TG-based estimation alone is insufficient for horizontal accelerometer biases, combining it with altitude-consistency constraints and nonlinear optimization enables stable bias determination under realistic operating conditions. For gyroscopes and the Z-axis accelerometer, the TG method remains effective, although additional analysis is required to fully explain the method-dependent variations observed in the experimental data.

## 5. Conclusions

This study confirmed that the TG-based bias-estimation framework can be successfully applied to real maritime IMU data. Among the six techniques evaluated, a low-pass Butterworth filter consistently yielded the most accurate accelerometer bias estimates, whereas the cross-correlation method performed poorly. For gyroscopes, most methods accurately estimated biases, suggesting that simple statistical approaches are sufficient for angular-rate sensors. The practical value of the estimated biases was further verified through simulations, which demonstrated their effectiveness in reducing navigation errors.

However, the TG approach alone was insufficient for determining static biases in the X- and Y-axis accelerometers, as these effects are indistinguishable from attitude errors caused by gravity [13]. To address this limitation, an altitude-based calibration was employed. By minimizing the difference between the vessel’s IGG-integrated altitude and the TG-computed altitude, we successfully inferred the horizontal biases, obtaining stable estimates of approximately 0.0405 m/s^2^ (X-axis) and 0.1615 m/s^2^ (Y-axis). These values were significantly larger than those observed in our previous simulation (≈0.022 and 0.020 m/s^2^, respectively), reflecting the influence of real sensor drift and environmental factors. Nonlinear optimization was crucial for refining these estimates and mitigating the “ping-pong” instability that occurred with a simpler iterative approach.

Nevertheless, it is important to note a key limitation of this study: the absence of ground truth validation for the sensor biases. Unlike controlled laboratory settings with precision equipment such as turntables, shipboard experiments do not allow for the direct measurement of true bias values. Consequently, the evaluation of estimation accuracy necessarily relied on indirect metrics, such as the statistical properties of the residuals and verification within a simulation environment. Therefore, future work should include sensitivity analyses in simulation and further experiments under diverse conditions to better quantify the effects of filter design, lever-arm errors, and environmental disturbances.

It should be noted that the accuracy of the trajectory generator (TG) used in this study was not directly verified with a high-grade reference system. While the TG output was compared with RTK-GNSS positional data to ensure general consistency, this approach provides only an indirect validation. A more rigorous assessment of TG accuracy—ideally involving comparisons with higher-precision navigation sensors such as fiber-optic or ring laser gyroscopes—would enable a more quantitative evaluation of the bias estimation performance. This remains an important topic for future work.

Overall, this study demonstrated that the combined framework is effective under realistic maritime conditions. The TG-based method with Butterworth filtering provides reliable estimates for gyroscope and Z-axis accelerometer biases, whereas altitude-difference calibration is essential for resolving X- and Y-axis accelerometer biases. This hybrid approach offers a practical solution for correcting biases in low-cost IMUs on vessels. Future work will focus on extending this validation to larger datasets, more diverse operational conditions, and other classes of low-cost navigation sensors.

## Figures and Tables

**Figure 1 sensors-25-06804-f001:**
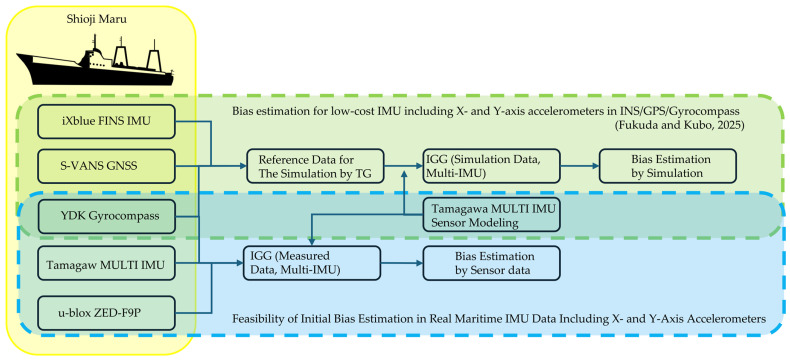
Schematic comparing the workflow of the previous simulation-based framework with that of the current experimental validation. Green blocks represent procedures from the previous study, whereas light-blue blocks indicate the new steps applied in this work. The areas enclosed in yellow represent the onboard equipment installed on the vessel. Among them, the green-shaded parts show the sensors and data processing used in [13], while the blue-shaded parts indicate the sensors and data processing employed in the present study (partly adapted from [13]).

**Figure 2 sensors-25-06804-f002:**
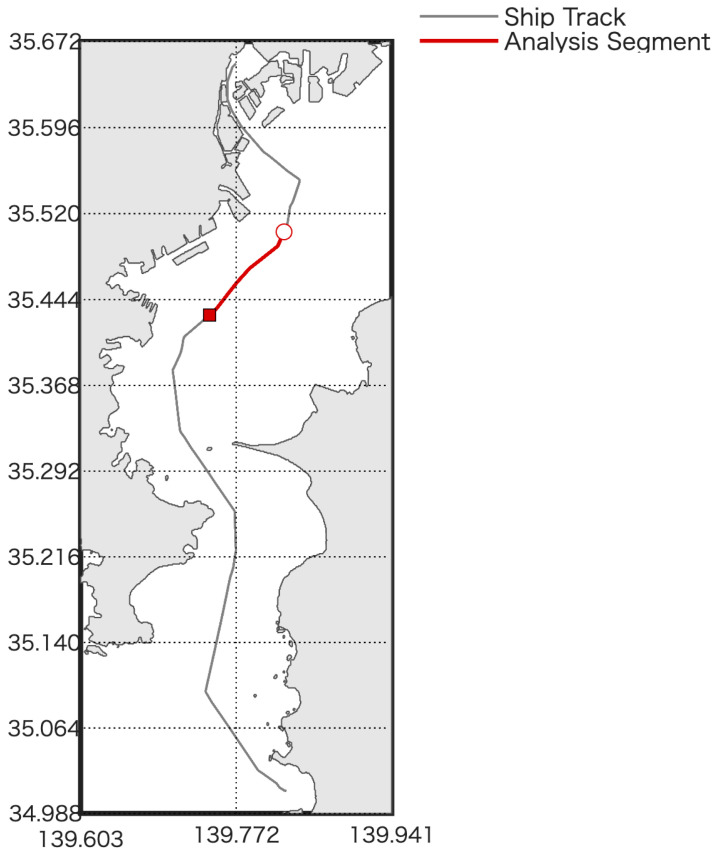
Experimental ship’s track in Tokyo Bay. The red line indicates the specific data segment used for the bias-estimation analysis. The gray line represents the actual trajectory of the vessel, while the red segment indicates the portion used for TG estimation.

**Figure 3 sensors-25-06804-f003:**
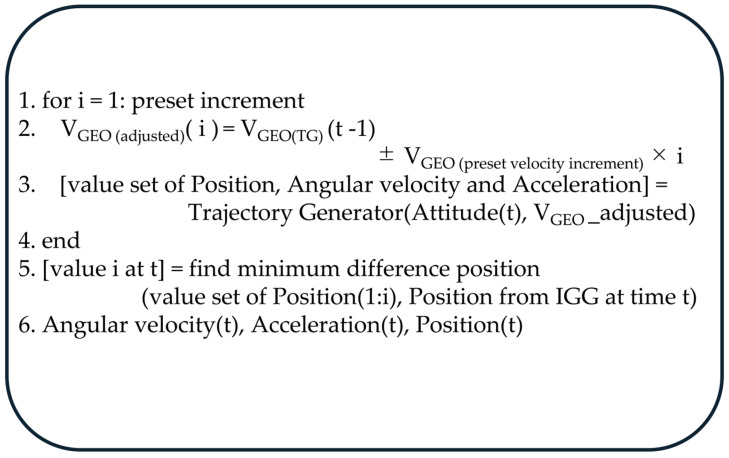
Conceptual flow of TG-Based Angular velocity and Acceleration estimation.

**Figure 4 sensors-25-06804-f004:**
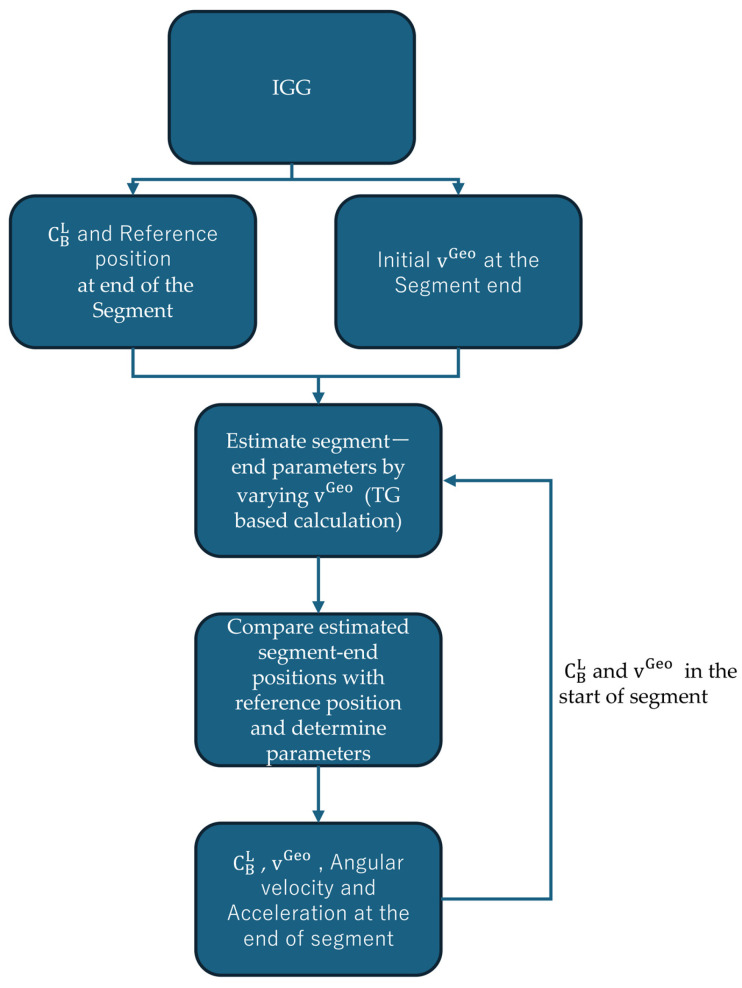
Processing flow of TG-based estimation within one segment.

**Figure 5 sensors-25-06804-f005:**
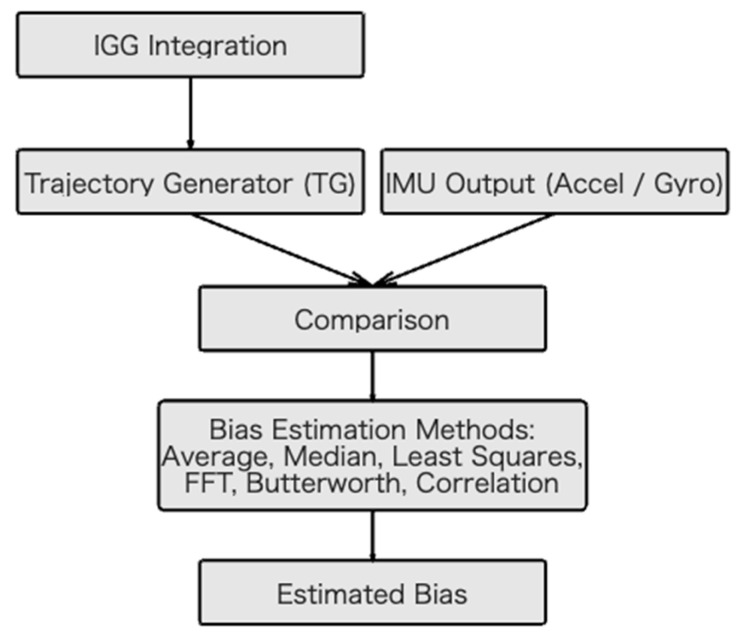
Block diagram illustrating the TG-based bias-estimation workflow.

**Figure 6 sensors-25-06804-f006:**
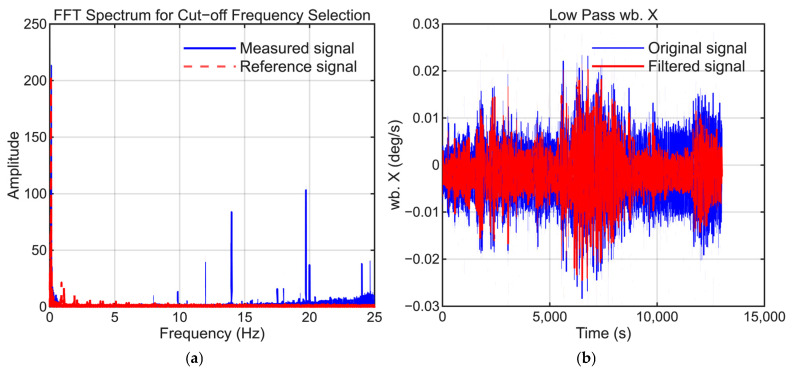
Data processing for the X-axis gyroscope (GyroX). (**a**) Fast Fourier transform (FFT) spectrum used to select the filter cutoff frequency. (**b**) Time-domain comparison of the raw sensor signal and that obtained after applying a low-pass filter.

**Figure 7 sensors-25-06804-f007:**
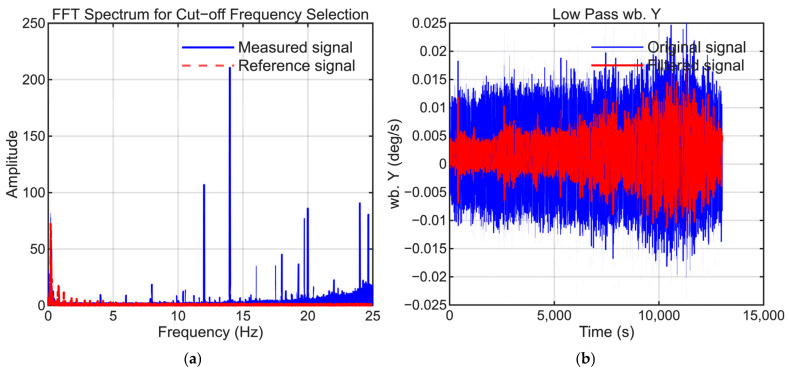
Data processing for the Y-axis gyroscope (GyroY). (**a**) FFT spectrum used to select the filter cutoff frequency. (**b**) Time-domain comparison of the raw and filtered signals.

**Figure 8 sensors-25-06804-f008:**
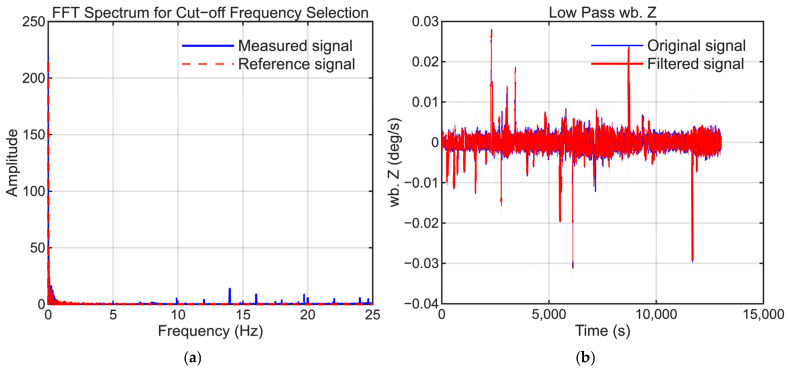
Data processing for the Z-axis gyroscope (GyroZ). (**a**) FFT spectrum used to select the filter cutoff frequency. (**b**) Time-domain comparison of the raw and filtered signals.

**Figure 9 sensors-25-06804-f009:**
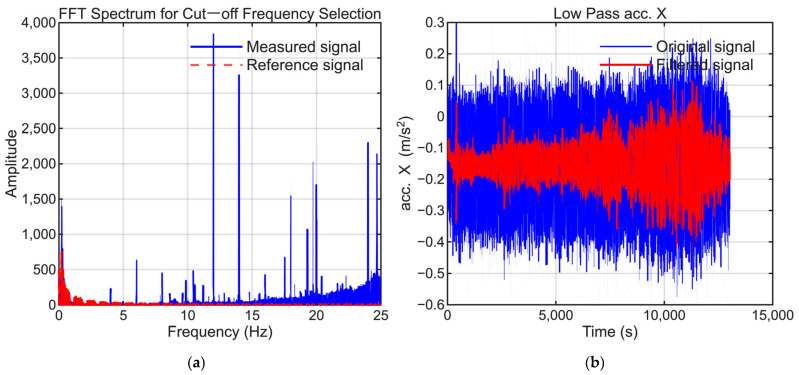
Data processing for the X-axis accelerometer (AccX). (**a**) FFT spectrum used to select the filter cutoff frequency. (**b**) Time-domain comparison of the raw and filtered signals.

**Figure 10 sensors-25-06804-f010:**
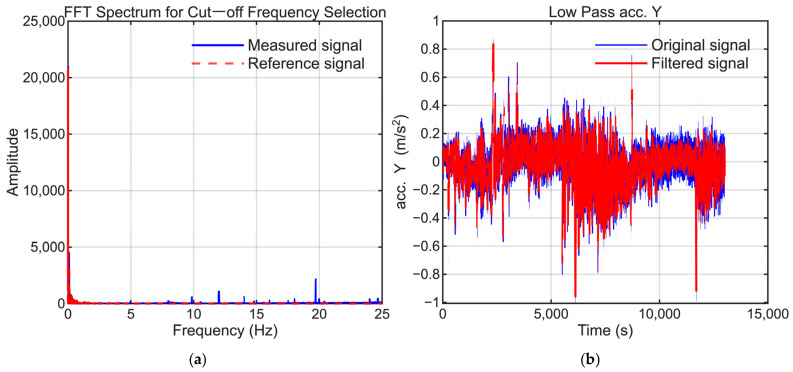
Data processing for the Y-axis accelerometer (AccY). (**a**) FFT spectrum used to select the filter cutoff frequency. (**b**) Time-domain comparison of the raw and filtered signals.

**Figure 11 sensors-25-06804-f011:**
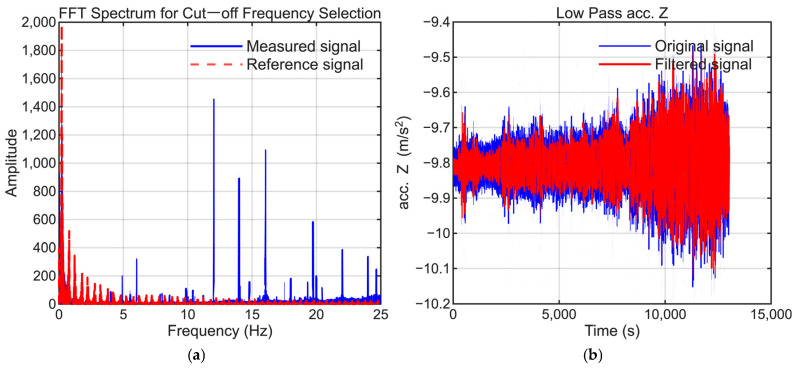
Data processing for the Z-axis accelerometer (AccZ). (**a**) FFT spectrum used to select the filter cutoff frequency. (**b**) Time-domain comparison of the raw and filtered signals.

**Figure 12 sensors-25-06804-f012:**
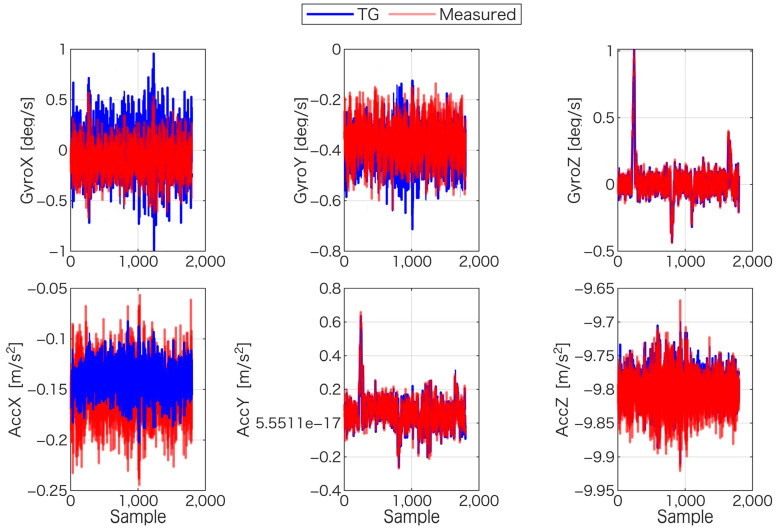
Comparison of bias-corrected IMU measurements (red) and the reference signals generated by the TG (blue) for all six sensor axes.

**Figure 13 sensors-25-06804-f013:**
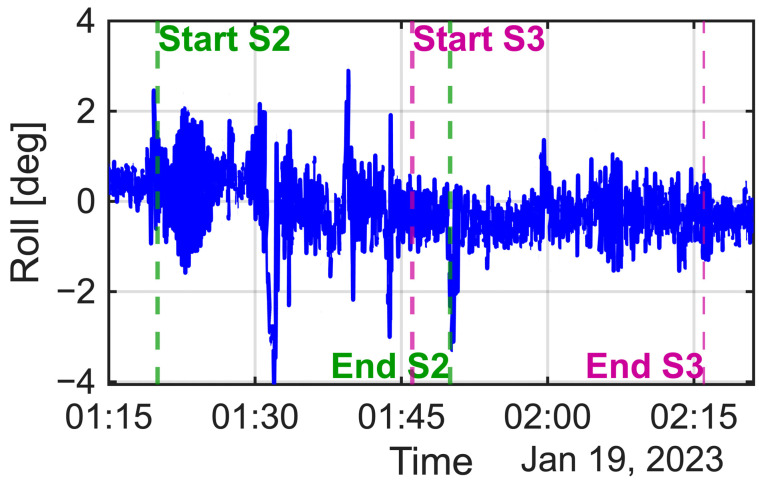
Time series of the vessel’s roll angle during the experiment, with the analysis intervals for Segment 2 (S2) and Segment 3 (S3) indicated.

**Figure 14 sensors-25-06804-f014:**
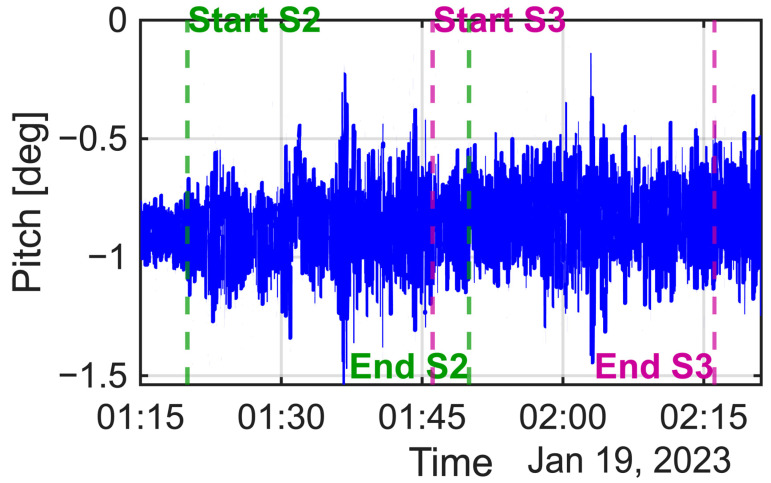
Time series of the vessel’s pitch angle during the experiment, with the analysis intervals for Segment 2 (S2) and Segment 3 (S3) indicated.

**Table 1 sensors-25-06804-t001:** Parameter estimation using AV [13].

	STD (deg/s) (m/s^2^)	Random Walk (deg/s$\sqrt{Hz}$) (m/s^2^$\sqrt{Hz}$)	**Bias Instability** **(deg/s) (m/s^2^)**
Gyro X	4.485 × 10^−2^	6.423 × 10^−3^	5.334 × 10^−3^
Gyro Y	3.998 × 10^−2^	5.968 × 10^−3^	4.922 × 10^−3^
Gyro Z	1.371 × 10^−3^	1.882 × 10^−4^	2.191 × 10^−5^
Acc X	1.920 × 10^−3^	2.063 × 10^−4^	2.534 × 10^−5^
Acc Y	2.412 × 10^−3^	2.958 × 10^−4^	3.073 × 10^−5^
Acc Z	2.526 × 10^−3^	3.013 × 10^−4^	7.848 × 10^−5^

**Table 2 sensors-25-06804-t002:** Summary of estimated sensor biases for each axis using the six different estimation methods.

Method	GyroX	GyroY	GyroZ	AccX	AccY	AccZ
	deg/s	deg/s	deg/s	m/s^2^	m/s^2^	m/s^2^
Average	−0.0913	0.1020	0.003585	−0.01033	0.00308	−0.01033
Normalized Correlation	−0.0900	−0.3000	0.000500	0.20000	−0.00492	0.20000
Least Squares	−0.0913	0.1020	0.003585	−0.01033	0.00308	−0.01033
FFT	−0.0913	0.1020	0.003585	−0.01033	0.00308	−0.01033
Butterworth	−0.0915	0.1017	0.003550	−0.01024	0.00308	−0.01034
Median	−0.0941	0.0998	0.004397	−0.00928	0.00493	−0.00956

**Table 3 sensors-25-06804-t003:** Mean residuals for each sensor axis and estimation method.

Sensor	Average	Corr	Least	FFT	Butter	Median
GyroX [deg/s]	0	−2.28 × 10^−5^	0	0	3.04 × 10^−6^	3.04 × 10^−6^
GyroY [deg/s]	0	7.02 × 10^−3^	0	0	5.89 × 10^−6^	3.89 × 10^−5^
GyroZ [deg/s]	0	5.38 × 10^−5^	0	0	6.12 × 10^−7^	−1.42 × 10^−5^
AccX [m/s^2^]	0	−2.10 × 10^−1^	0	0	−8.50 × 10^−5^	−1.05 × 10^−3^
AccY [m/s^2^]	0	8.00 × 10^−3^	0	0	−5.94 × 10^−6^	−1.85 × 10^−3^
AccZ [m/s^2^]	0	−2.10 × 10^−1^	0	0	9.80 × 10^−6^	−7.74 × 10^−4^

**Table 4 sensors-25-06804-t004:** Median residuals for each sensor axis and estimation method.

Sensor	Average	Corr	Least	FFT	Butter	Median
GyroX [deg/s]	−4.93 × 10^−5^	−7.21× 10^−5^	−4.93 × 10^−5^	−4.93 × 10^−5^	−4.63 × 10^−5^	−4.63 × 10^−5^
GyroY [deg/s]	−3.89 × 10^−5^	6.98 × 10^−3^	−3.89 × 10^−5^	−3.89 × 10^−5^	−3.30 × 10^−5^	0
GyroZ [deg/s]	1.42 × 10^−5^	6.80 × 10^−5^	1.42 × 10^−5^	1.42 × 10^−5^	1.48 × 10^−5^	0
AccX [m/s^2^]	1.05 × 10^−3^	−2.09 × 10^−1^	1.05 × 10^−3^	1.05 × 10^−3^	9.61 × 10^−4^	0
AccY [m/s^2^]	1.85 × 10^−3^	9.85 × 10^−3^	1.85 × 10^−3^	1.85 × 10^−3^	1.85 × 10^−3^	0
AccZ [m/s^2^]	7.74 × 10^−4^	−2.10 × 10^−1^	7.74 × 10^−4^	7.74 × 10^−4^	7.84 × 10^−4^	0

**Table 5 sensors-25-06804-t005:** RMS residuals for each sensor axis and estimation method.

Sensor	Average	Corr	Least	FFT	Butter	Median
GyroX [deg/s]	3.28 × 10^−3^	3.28 × 10^−3^	3.28 × 10^−3^	3.28 × 10^−3^	3.28 × 10^−3^	3.28 × 10^−3^
GyroY [deg/s]	3.49 × 10^−3^	7.84 × 10^−3^	3.49 × 10^−3^	3.49 × 10^−3^	3.49 × 10^−3^	3.49 × 10^−3^
GyroZ [deg/s]	5.58 × 10^−4^	5.61 × 10^−4^	5.58 × 10^−4^	5.58 × 10^−4^	5.58 × 10^−4^	5.58 × 10^−4^
AccX [m/s^2^]	3.04 × 10^−2^	2.13 × 10^−1^	3.04 × 10^−2^	3.04 × 10^−2^	3.04 × 10^−2^	3.05 × 10^−2^
AccY [m/s^2^]	3.80 × 10^−2^	3.88 × 10^−2^	3.80 × 10^−2^	3.80 × 10^−2^	3.80 × 10^−2^	3.80 × 10^−2^
AccZ [m/s^2^]	2.30 × 10^−2^	2.12 × 10^−1^	2.30 × 10^−2^	2.30 × 10^−2^	2.30 × 10^−2^	2.30 × 10^−2^

**Table 6 sensors-25-06804-t006:** Maximum normalized cross-correlation coefficients for each sensor axis and estimation method.

Sensor	Average	Corr	Least	FFT	Butter	Median
GyroX	0.783	0.783	0.783	0.783	0.783	0.783
GyroY	0.638	0.126	0.638	0.638	0.638	0.637
GyroZ	0.971	0.971	0.971	0.971	0.971	0.971
AccX	0.986	0.995	0.986	0.986	0.986	0.986
AccY	0.953	0.954	0.953	0.953	0.953	0.952
AccZ	1.00	1.00	1.00	1.00	1.00	1.00

**Table 7 sensors-25-06804-t007:** Simulation results for gyroscope bias estimation. True bias values were set based on the average/least squares results from the experimental data (Table 2).

Method	X Gyro deg/s		Y Gyro deg/s		Z Gyro deg/s	
	−0.0913		0.102		0.003585	
	Est.	Error	Est.	Error	Est.	Error
Average	−0.0914	0.0001	0.0995	0.0025	0.003580	0.000005
Normalized correlation	−0.0900	0.0013	0.1000	0.0020	0.000500	0.003085
Least squares	−0.0914	0.0001	0.0995	0.0025	0.003580	0.000005
FFT	−0.0914	0.0001	0.0995	0.0025	0.003580	0.000005
Butterworth	−0.0914	0.0001	0.0997	0.0023	0.003586	0.000001
Median	−0.0925	0.0012	0.1013	0.0007	0.003671	0.000086

**Table 8 sensors-25-06804-t008:** Simulation results for accelerometer bias estimation. True bias values were set based on the experimental results from the average/least squares (Table 1) and altitude-based methods.

Method	X Acc m/s^2^		Y Acc m/s^2^		Z Acc m/s^2^	
	0.03900		0.01600		−0.01033	
	Est.	Error	Est.	Error	Est	Error
Average	−5.8223 × 10^−3^	4.4822 × 10^−2^	−2.4506 × 10^−3^	1.8451 × 10^−2^	−1.0186 × 10^−2^	1.4413 × 10^−4^
Normalized correlation	−4.4700 × 10^−3^	4.3470 × 10^−2^	−2.5200 × 10^−3^	1.8520 × 10^−2^	1.0731 × 10^−1^	1.1764 × 10^−1^
Least squares	−5.8223 × 10^−3^	4.4822 × 10^−2^	−2.4506 × 10^−3^	1.8451 × 10^−2^	−1.0186 × 10^−2^	1.4413 × 10^−4^
FFT	−5.8223 × 10^−3^	4.4822 × 10^−2^	−2.4506 × 10^−3^	1.8451 × 10^−2^	−1.0186 × 10^−2^	1.4413 × 10^−4^
Butterworth	−5.8532 × 10^−3^	4.4853 × 10^−2^	−2.4028 × 10^−3^	1.8403 × 10^−2^	−1.0183 × 10^−2^	1.4737 × 10^−4^
Median	−6.0291 × 10^−3^	4.5029 × 10^−2^	−2.4424 × 10^−3^	1.8442 × 10^−2^	−1.0285 × 10^−2^	4.4625 × 10^−5^

**Table 9 sensors-25-06804-t009:** Simulation results for gyroscope bias estimation. True bias values were set based on the median-method results from the experimental data (Table 1).

Method	X Gyro deg/s		Y Gyro deg/s		Z Gyro deg/s	
	−0.09413		0.0998		0.004397	
	Est.	Error	Est.	Error	Est.	Error
Average	−0.0943	0.0001	0.0973	0.0025	0.004392	0.000005
Normalized correlation	−0.0900	0.0041	0.1000	0.0002	0.000500	0.003897
Least squares	−0.0943	0.0001	0.0973	0.0025	0.004392	0.000005
FFT	−0.0943	0.0001	0.0973	0.0025	0.004392	0.000005
Butterworth	−0.0943	0.0001	0.0975	0.0023	0.004397	0.000000
Median	−0.0953	0.0012	0.0991	0.0007	0.004482	0.000085

**Table 10 sensors-25-06804-t010:** Simulation results for accelerometer bias estimation. True bias values were set based on the median (Table 1) and altitude-based methods.

Method	X Acc m/s^2^		Y Acc m/s^2^		Z Acc m/s^2^	
	0.03900		0.01600		−0.00956	
	Est.	Error	Est.	Error	Est	Error
Average	−5.7525 × 10^−3^	4.4753 × 10^−2^	−2.5897 × 10^−3^	1.8590 × 10^−2^	−9.4128 × 10^−3^	1.4522 × 10^−4^
Normalized correlation	−4.4100 × 10^−3^	4.3410 × 10^−2^	−2.6600 × 10^−3^	1.8660 × 10^−2^	1.0799 × 10^−1^	1.1755 × 10^−1^
Least squares	−5.7525 × 10^−3^	4.4753 × 10^−2^	−2.5897 × 10^−3^	1.8590 × 10^−2^	−9.4128 × 10^−3^	1.4522 × 10^−4^
FFT	−5.7525 × 10^−3^	4.4753 × 10^−2^	−2.5897 × 10^−3^	1.8590 × 10^−2^	−9.4128 × 10^−3^	1.4522 × 10^−4^
Butterworth	−5.7835 × 10^−3^	4.4783 × 10^−2^	−2.5418 × 10^−3^	1.8542 × 10^−2^	−9.4095 × 10^−3^	1.4846 × 10^−4^
Median	−6.0573 × 10^−3^	4.5057 × 10^−2^	−2.6254 × 10^−3^	1.8625 × 10^−2^	−9.5176 × 10^−3^	4.0386 × 10^−5^

**Table 11 sensors-25-06804-t011:** Estimation results obtained in Segment 3.

For Loop	First		Second		Third		Fourth	
X-axis bias m/s^2^	0.05	(0.05)	0.04	(0.04)	0.04	(0.04)	0.04	(0.04)
Y-axis bias m/s^2^	0.00	0.15	(0.15)	0.16	(0.16)	0.16	(0.16)	0.16

## Data Availability

The data presented in this study are available upon request from the corresponding author. These data are not publicly available because of the training ship. When a request is received for a reasonable reason, it can only be provided after explaining it to the relevant department and obtaining permission from all the departments.

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
