# Peer review of "Feasibility of Initial Bias Estimation in Real Maritime IMU Data Including X- and Y-Axis Accelerometers"

_sensors, 2025, doi:10.3390/s25216804_

Round 1

Reviewer 1 Report

Comments and Suggestions for Authors

This paper uses Extended Kalman Filter (EKF) to fuse Global Navigation Satellite System (GNSS), Inertial Navigation System (INS), and gyro compass data to construct a Trajectory Generator (TG) to generate unbiased reference signals. This trajectory generator is used to estimate the deviation of gyroscopes and Z-axis accelerometers, solving the problem of X-axis and Y-axis accelerometer deviation that cannot be directly observed.

However, the paper still has the following shortcomings:

  1. As a continuation of the previous paper (Bias Estimation for Low-Cost IMU Including X- and Y-Axis Accelerometers in INS/GPS/Gyrocompass), this paper only briefly mentions the simulation results of the previous research in the introduction, without systematically summarizing the continuity and differences in the methodological systems of the two papers. Suggest adding a section on "Method Inheritance and Innovation" to clarify the basis for using and adjusting the core parameters of the TG framework in the previous paper for real data validation in this paper. At the same time, compare the similarities and differences in the X/Y axis accelerometer deviation estimation process between the two papers to enhance the coherence and logicality of the research.
  2. Both papers use "minimizing the height difference between IGG and TG" to estimate the X/Y axis accelerometer deviation, but this article does not attempt to explain the theoretical correlation between height difference and accelerometer deviation as in the previous paper, and only verifies its effectiveness through experimental phenomena. Suggest adding a theoretical derivation section, combining INS navigation equations (such as altitude channel dynamics models), to derive the mathematical relationship between X/Y-axis accelerometer deviation and altitude estimation indirectly affected by attitude angles (roll, pitch), and clarify the quantitative correlation between altitude difference and deviation.
  3. Section 3.1.3 of the previous paper mentions the use of 1Hz low-pass filtering for X-axis acceleration and 5Hz low-pass filtering for Y/Z axis. Although the same filtering strategy is used in Section 2.3 of this paper, it does not explain the verification process of the adaptability of the filtering parameters in real maritime environments (such as ship vibration frequency under moderate sea conditions). It is suggested to supplement the sensitivity analysis of different filtering parameters on the deviation estimation results, prove the rationality of the current parameter selection, and explain the specific frequency of periodic noise in real data and the design details of notch filters (such as center frequency and bandwidth), in order to enhance the reproducibility of data processing.
  4. Incomplete comparison design for simulation verification: Section 3.2 of this paper verifies the deviation of real data estimation based on the simulation environment of the previous paper, but it is not clear whether the reference data from Section 3.1.1 of the previous paper will be used in the simulation, nor is there a comparison between the "deviation of real data estimation" and the "optimal deviation of the previous paper simulation" in terms of navigation error suppression effect. Suggest adding two sets of control experiments: one is to simulate the navigation results of bias using the previous paper, and the other is to estimate the navigation results of bias using the real data in this article, thus demonstrating the value of verifying the real data in this paper.
  5. Insufficient description of the details of the experimental environment: Section 2.1 of this paper only mentions "moderate sea conditions" and "regular navigation for 30 minutes", without supplementing the specific navigation trajectory shown in Figure 6 of the preceding paper (such as latitude and longitude range, number of turns), environmental interference factors during the experiment, etc. It is recommended to supplement experimental log related information, including specific quantitative indicators of sea conditions and the integrity of GNSS data. These information are crucial for evaluating the reliability of experimental data and the applicable boundaries of methods.
  6. Normative issues with references and formats: In this article, the citation format of reference [8] is inconsistent with other references (such as missing DOI numbers). It is recommended to standardize the reference format, strictly follow the citation standards of MDPI's "Sensors" journal, and supplement the complete access dates and specific page information of all online literature; At the same time, there is a misalignment in the unit annotation of the table (such as Table 1) ("deg/s m/s²" does not correspond to a specific column), and the table format needs to be corrected to ensure that the units correspond one-to-one with the parameters, thereby improving the standardization and readability of the paper.

Author Response

Comments 1: This paper uses Extended Kalman Filter (EKF) to fuse Global Navigation Satellite System (GNSS), Inertial Navigation System (INS), and gyro compass data to construct a Trajectory Generator (TG) to generate unbiased reference signals. This trajectory generator is used to estimate the deviation of gyroscopes and Z-axis accelerometers, solving the problem of X-axis and Y-axis accelerometer deviation that cannot be directly observed.

Response 1: We sincerely thank the reviewer for their valuable feedback and constructive suggestions, which have greatly contributed to improving the overall quality of our manuscript. We have carefully addressed all comments by thoroughly reviewing the paper and making the necessary revisions. Detailed, point-by-point responses are provided below, and the corresponding changes are highlighted in green in the revised manuscript for clarity.

As you recommended additional English editing, we have conducted a second round of professional proofreading by a certified academic editing service. Because this process affected the overall wording throughout the manuscript, it was not feasible to highlight every individual change. Therefore, we have also included a version of the revised manuscript with Microsoft Word’s “Track Changes” function enabled, so that all textual modifications can be clearly identified.

Once again, we sincerely appreciate the reviewer’s insightful comments, which have helped us improve the clarity, structure, and completeness of our paper.

However, the paper still has the following shortcomings:

Comments 2: As a continuation of the previous paper (Bias Estimation for Low-Cost IMU Including X- and Y-Axis Accelerometers in INS/GPS/Gyrocompass), this paper only briefly mentions the simulation results of the previous research in the introduction, without systematically summarizing the continuity and differences in the methodological systems of the two papers. Suggest adding a section on "Method Inheritance and Innovation" to clarify the basis for using and adjusting the core parameters of the TG framework in the previous paper for real data validation in this paper. At the same time, compare the similarities and differences in the X/Y axis accelerometer deviation estimation process between the two papers to enhance the coherence and logicality of the research.

Response 2: Thank you very much for your valuable comment. The main purpose of the present paper is to examine how the bias estimation framework, which was validated in our previous simulation-based study, performs when applied to real measurement data. For this reason, we employed the same methodological system as in the simulation study.

For the X- and Y-axis accelerometers, however, a “ping-pong” phenomenon was observed in the real data, which had not appeared in the simulation. To address this, we newly introduced a nonlinear optimization approach. Other aspects of the TG configuration, Kalman filter parameters, and framework settings remain consistent with those in the simulation, except that the TG is now applied only to the analyzed data segment.

In response to your suggestion, we have added a new subsection entitled “1.1 Method Inheritance and Innovation”to explicitly describe the continuity and differences between the two studies. Additionally, we included Figure 1 to illustrate the relationship between the previous research and the present study.

We believe these revisions improve the coherence and logical consistency of the manuscript, as you recommended.

Comments 3: Both papers use "minimizing the height difference between IGG and TG" to estimate the X/Y axis accelerometer deviation, but this article does not attempt to explain the theoretical correlation between height difference and accelerometer deviation as in the previous paper, and only verifies its effectiveness through experimental phenomena. Suggest adding a theoretical derivation section, combining INS navigation equations (such as altitude channel dynamics models), to derive the mathematical relationship between X/Y-axis accelerometer deviation and altitude estimation indirectly affected by attitude angles (roll, pitch), and clarify the quantitative correlation between altitude difference and deviation.

Response 3: We sincerely appreciate the reviewer’s valuable comment. In this study, we aimed to verify whether the X- and Y-axis accelerometer biases could be estimated from real measurement data using the same methodology as in our previous simulation-based work. The reason why these biases affect altitude estimation is that even a small tilt error in roll and pitch angles can lead to a slight error in the vertical acceleration component, which accumulates over time and manifests as an altitude difference.
However, as also discussed in our previous study, it is extremely challenging to derive a closed-form theoretical expression that quantitatively describes how the accelerometer bias propagates through the roll/pitch dynamics and affects the altitude estimate in the INS solution. Simulation results demonstrated that this influence appears on the order of 10⁻² m, and the primary purpose of this study was to investigate whether such small effects could still be captured and estimated under real-world conditions.
We agree that establishing a rigorous mathematical model of this relationship would further enhance the estimation framework, and this is an important direction for future work. Nevertheless, we believe that demonstrating the feasibility of bias estimation from real measurements — consistent with the trends observed in simulations — is a significant contribution of this paper.
Furthermore, we have revised Section 2.2.3 to incorporate an additional explanation addressing the reviewer’s suggestion. In this revision, we qualitatively discuss how small X- and Y-axis accelerometer biases can affect altitude estimation through their coupling with roll and pitch dynamics, and we explain that the resulting vertical acceleration error can accumulate and manifest as altitude deviations. Although deriving a closed-form analytical model remains challenging, this added discussion clarifies the theoretical background of the bias–altitude relationship as requested.

Comments 4: Section 3.1.3 of the previous paper mentions the use of 1Hz low-pass filtering for X-axis acceleration and 5Hz low-pass filtering for Y/Z axis. Although the same filtering strategy is used in Section 2.3 of this paper, it does not explain the verification process of the adaptability of the filtering parameters in real maritime environments (such as ship vibration frequency under moderate sea conditions). It is suggested to supplement the sensitivity analysis of different filtering parameters on the deviation estimation results, prove the rationality of the current parameter selection, and explain the specific frequency of periodic noise in real data and the design details of notch filters (such as center frequency and bandwidth), in order to enhance the reproducibility of data processing.

Response 4: We sincerely appreciate the reviewer’s insightful comment regarding the verification of filtering parameter adaptability and sensitivity analysis.
In this study, the primary objective is to verify how the bias estimation method, which was validated in simulations, performs when applied to real measurement data. Because the true values of sensor biases are unknown in real-world experiments, the cutoff frequencies of the low-pass filters were determined empirically by comparing the frequency characteristics of the TG reference signals and the measured IMU signals (we have now included the FFT comparison figure to clarify this point).
We fully agree that conducting a sensitivity analysis on how different filtering parameters influence bias estimation results is an important step. However, since this paper focuses specifically on validating simulation-derived findings with real-world data, we have intentionally kept the filtering design simple, relying on the TG–IMU frequency comparison and empirical knowledge. Additionally, factors such as lever-arm effects—which were not addressed even in the simulation study—could affect the reproducibility of results. We believe that parameter tuning without a comprehensive simulation-based investigation might risk overfitting to a specific dataset.
Therefore, as part of future work, we plan to systematically investigate the influence of filtering parameter selection within a simulation framework and subsequently apply the findings to real-world data analysis.

Comments 5: Incomplete comparison design for simulation verification: Section 3.2 of this paper verifies the deviation of real data estimation based on the simulation environment of the previous paper, but it is not clear whether the 

Response 5: Thank you very much for your insightful comment, and we sincerely apologize for any lack of clarity in our explanation. As illustrated in the newly added Figure 1, the real data and the simulation data were obtained from the same experimental campaign, with the only difference being the type of sensor used. Therefore, the navigation conditions and trajectories are exactly the same.
In the simulation, the bias values estimated from the real data in this study were applied, while all other conditions were kept identical to those in the previous paper. Thus, the purpose of the simulation in this paper was not to re-perform the same navigation test, but to specifically examine whether the magnitude of the estimated biases influences the stability of the estimation results.
As you correctly pointed out, the real data estimation results exhibited some variability. However, when the same bias values were applied in the simulation, the results were consistent with those obtained in our previous study. This indicates that the variability observed in the present study arises from the real measurement data (e.g., modeling errors, lever-arm correction uncertainties, or environmental disturbances), rather than from the estimation method itself.
For this reason, we believe that an additional control experiment is not necessary, and that the current comparison sufficiently demonstrates the value of verifying the TG-based bias estimation framework using real measurement data, which is the main contribution of this paper.

Comments 6: Insufficient description of the details of the experimental environment: Section 2.1 of this paper only mentions "moderate sea conditions" and "regular navigation for 30 minutes", without supplementing the specific navigation trajectory shown in Figure 6 of the preceding paper (such as latitude and longitude range, number of turns), environmental interference factors during the experiment, etc. It is recommended to supplement experimental log related information, including specific quantitative indicators of sea conditions and the integrity of GNSS data. These information are crucial for evaluating the reliability of experimental data and the applicable boundaries of methods.

Response 6: Thank you for your valuable comment. We apologize for the insufficient explanation in the original manuscript.
As clarified in the newly added Figure 1, the real data used in this study were obtained from the same experimental campaign as in our previous work [13]. Therefore, the navigation trajectory (including latitude/longitude range and number of turns) is identical to that presented in Figure 6 of the previous paper. The only difference lies in the type of sensor used.
We believe that, with the addition of Figure 1 and this clarification, readers can better understand the experimental setting and the continuity between the two studies. We have revised Section 2.1 to reflect this explanation.

Comments 7: Normative issues with references and formats: In this article, the citation format of reference [8] is inconsistent with other references (such as missing DOI numbers). It is recommended to standardize the reference format, strictly follow the citation standards of MDPI's "Sensors" journal, and supplement the complete access dates and specific page information of all online literature; At the same time, there is a misalignment in the unit annotation of the table (such as Table 1) ("deg/s m/s²" does not correspond to a specific column), and the table format needs to be corrected to ensure that the units correspond one-to-one with the parameters, thereby improving the standardization and readability of the paper.

Response 7: We sincerely thank the reviewer for pointing out the issues regarding reference formatting and table presentation.
We have carefully revised the references to comply with the MDPI Sensors citation style. In particular, the DOI number has been added to reference [8], and the access dates for online resources that were previously missing have now been included.
Regarding Table 1, the units “deg/s” and “m/s²” are applied to the gyroscope (Gyro) and accelerometer (Acc) columns, respectively. The table was originally formatted so that each unit corresponded to three columns (X, Y, and Z), and we believe the parameter–unit correspondence was technically correct. However, as the reviewer pointed out potential readability issues, we have improved the layout and clarified the correspondence between units and parameters.
For the other tables, we carefully reviewed them and found no clear misalignments. We believe they are understandable and consistent with the presentation style commonly found in similar research papers. 

Reviewer 2 Report

Comments and Suggestions for Authors

The presented manuscript is a direct continuation of the previous article by the authors (Fukuda, G.; Kubo, N. Bias estimation for low-cost IMU including X- and Y-axis accelerometers in INS/GPS/Gyrocompass. Sensors (Basel) 2025, 25, 1315. DOI:10.3390/s25051315.), which proposes an improved method for bias estimation by comparing the estimated values from a trajectory generator (TG)-based acceleration and angular-velocity estimation system with actual measurements. Additionally, for X- and Y-axis accelerations, the authors introduce a method that leverages the correlation between altitude differences derived from an INS/GNSS/gyrocompass (IGG) and those obtained during the TG estimation process to estimate the bias. The presented manuscript focuses on the validation of the method using real-world IMU data obtained in maritime experiments.

Six bias estimation approaches—including statistical estimators (mean, median), model-based (least squares, normalized cross-correlation), and signal-processing methods (FFT, Butterworth filter)—were compared using real maritime measurement data. As the true bias values were unknown, the validity of the estimation was assessed based on the statistical properties of the residuals (mean, median, root-mean-squared (RMS) residuals, and normalized cross-correlation).

Specific comments

  1. Even though the proposed manuscript is a direct continuation of the previous work of the authors, it should be an independent document. Now, to understand the material presented, it is necessary to read the previous work of the authors. It is necessary to add a description of the TG-based bias estimation method and its approaches in the main part of the paper.
  2. The abstract says "This study employed an Extended Kalman Filter (EKF) to integrate GNSS, INS, and gyrocompass data, forming a TG that produced bias-free reference signals." How TG works is not described either in this or in the previous article by the authors. Since the TG uses INS data, there may be errors in the generated signal (acceleration and angular rate) that correlate with the errors of the IMU. How are these errors eliminated?
  3. The information presented in the introduction describes the motivation of the authors' total research, but only indirectly relates to the problem considered in the paper. In the introduction, it is necessary to describe the problem being solved in more detail. It is necessary to indicate the advantages and differences of the proposed method from the existing ones.
  4. Bias Estimation Accuracy. The use of filters for signal processing can lead to the appearance of low-frequency components in the signal that are indistinguishable from biases. It is not enough to estimate the error only by analyzing the residuals. A visual way to demonstrate the effectiveness of the proposed methods would be to compare the accuracy of the generated INS position (or velocity) before and after biases are taken into account.
  5. It is necessary to specify the parameters of gyroscopes and accelerometers, particularly the noise level, to evaluate the bias estimate accuracy.
  6. The section "3.2 Verification Result in a Simulation Environment" does not describe the essence of the verification. It is unclear how the verification was carried out. There is no description of the methodology.
  7. The text says "The estimated biases were then applied in the simulation environment from our previous study [6] to examine their effect on navigation accuracy." However, there is no assessment of the effect of biases on navigation accuracy in the paper.
  8. The text " EKF was used to integrate INS, GNSS, and gyrocompass data, providing reference estimates of attitude, position, and velocity [2]" contains an incorrect reference.
  9. In the text of section 3.1 and Tables 2-4, 6,7, there are no units of measurement of the estimated values.

Author Response

Comments 1: The presented manuscript is a direct continuation of the previous article by the authors (Fukuda, G.; Kubo, N. Bias estimation for low-cost IMU including X- and Y-axis accelerometers in INS/GPS/Gyrocompass. Sensors (Basel) 2025, 25, 1315. DOI:10.3390/s25051315.), which proposes an improved method for bias estimation by comparing the estimated values from a trajectory generator (TG)-based acceleration and angular-velocity estimation system with actual measurements. Additionally, for X- and Y-axis accelerations, the authors introduce a method that leverages the correlation between altitude differences derived from an INS/GNSS/gyrocompass (IGG) and those obtained during the TG estimation process to estimate the bias. The presented manuscript focuses on the validation of the method using real-world IMU data obtained in maritime experiments.

Six bias estimation approaches—including statistical estimators (mean, median), model-based (least squares, normalized cross-correlation), and signal-processing methods (FFT, Butterworth filter)—were compared using real maritime measurement data. As the true bias values were unknown, the validity of the estimation was assessed based on the statistical properties of the residuals (mean, median, root-mean-squared (RMS) residuals, and normalized cross-correlation).

Response 1: We sincerely thank the reviewer for taking the time to evaluate our manuscript and for the constructive comments provided. Below, we address each point raised in detail.

Please note that all major revisions and clarifications have been **highlighted in green** in the revised manuscript for ease of review.

Additionally, as one of the reviewers previously recommended further English editing, we have requested a **second round of professional proofreading** by a certified academic editing service. Because this process affected the overall wording throughout the manuscript, it was not possible to highlight every individual change. Therefore, we have also provided a version of the revised manuscript with **Microsoft Word’s “Track Changes” function enabled**, so that all textual modifications can be clearly identified.

Once again, we thank you for your valuable feedback, which has helped us to improve the clarity, structure, and completeness of our paper.

Comments 2: Even though the proposed manuscript is a direct continuation of the previous work of the authors, it should be an independent document. Now, to understand the material presented, it is necessary to read the previous work of the authors. It is necessary to add a description of the TG-based bias estimation method and its approaches in the main part of the paper.

Response 2: Thank you for your valuable comment. We have added a detailed description of the TG-based bias estimation method and its approaches in Section 2.2.1 to make the manuscript more self-contained and understandable without referring to our previous work.

Comments 3: The abstract says "This study employed an Extended Kalman Filter (EKF) to integrate GNSS, INS, and gyrocompass data, forming a TG that produced bias-free reference signals." How TG works is not described either in this or in the previous article by the authors. Since the TG uses INS data, there may be errors in the generated signal (acceleration and angular rate) that correlate with the errors of the IMU. How are these errors eliminated?

Response 3: We sincerely thank the reviewer for this valuable comment and for pointing out the need to clarify how the Trajectory Generator (TG) produces bias-free reference signals.
As described in the revised manuscript and illustrated in the newly added Figure 3, the TG-generated accelerations and angular rates are computed based on the reference attitude, velocity, and position estimated by the IGG (INS/GNSS/Gyrocompass integration).
The velocity estimated by the IGG inherently includes small IMU and estimation errors. Therefore, if it were directly used as the TG input, these errors would propagate into the TG-estimated position. To mitigate this, the TG applies an iterative process in which the IGG velocity is adjusted by adding and subtracting a preset velocity increment (Δv) around the nominal IGG-estimated velocity. For each adjusted velocity, the TG computes a dataset of position, angular rate, and acceleration values. Among these datasets, the one whose resulting TG-estimated position is closest to the IGG reference position is selected as the representative value at time t.
The accelerations and angular rates associated with this selected dataset correspond to the inputs that yield zero position error when used in inertial navigation computation. Hence, they are regarded as bias-free reference signals.
In other words, although the IGG reference position may still include minor IMU and GNSS errors, the TG-derived acceleration and angular rate values that reproduce this position without residual error are, by definition, free from bias with respect to the navigation outcome.
However, for the X- and Y-axis accelerometers, biases appear as constant components in the roll and pitch angles estimated by the IGG. These are not fully separable within the TG process itself and therefore require an additional estimation step, as described later in the manuscript.

Comments 4: The information presented in the introduction describes the motivation of the authors' total research, but only indirectly relates to the problem considered in the paper. In the introduction, it is necessary to describe the problem being solved in more detail. It is necessary to indicate the advantages and differences of the proposed method from the existing ones.

Response 4: We sincerely thank the reviewer for this helpful comment. As you correctly noted, the original introduction focused primarily on the overall motivation of our research series, which may have made the specific problem addressed in this paper less explicit.
In response, and in line with similar feedback from another reviewer, we have revised the Introduction and added a new subsection, *1.2 Method Inheritance and Innovation*. This new section clearly defines the scope and unique contribution of the present study compared to our previous work. It explicitly describes the specific problem solved in this paper—validating the bias estimation framework using real-world maritime IMU data—and clarifies the advantages and methodological differences of the proposed approach over existing methods.

These revisions make the structure and focus of the paper clearer to readers, helping to distinguish this work as an independent and self-contained contribution within the broader research context.

Comments 5: Bias Estimation Accuracy. The use of filters for signal processing can lead to the appearance of low-frequency components in the signal that are indistinguishable from biases. It is not enough to estimate the error only by analyzing the residuals. A visual way to demonstrate the effectiveness of the proposed methods would be to compare the accuracy of the generated INS position (or velocity) before and after biases are taken into account.

Response 5: We sincerely thank the reviewer for this important and insightful comment.
We considered performing a comparison using a pure INS (without external aiding) to directly demonstrate the impact of bias estimation on navigation accuracy. However, due to the limited precision of the MEMS-based IMU used in this study, such a comparison was found to be impractical. Specifically, when operating in a pure INS mode, the positional error typically diverges by tens to hundreds of meters within just a few seconds, making it extremely difficult to distinguish whether the resulting drift originates from the estimated bias, the low-pass filter configuration, unmodeled vibrations, or other sensor errors.
For this reason, our previous study first validated the proposed bias estimation framework through simulation analysis, where the true bias values and noise characteristics were fully controlled. That work demonstrated the theoretical effectiveness of the proposed approach. The present paper, in turn, focuses on analyzing how the same framework performs under real-world experimental conditions, where various sources of error coexist.
As the reviewer correctly notes, MEMS accelerometers exhibit inherently poor precision compared with navigation-grade sensors. Therefore, in our shipborne experiments, we rely on external aiding such as Doppler Velocity Logs (DVLs) for accurate velocity estimation.(https://doi.org/10.3390/s21041056) Although a pure INS comparison would ideally be the most straightforward validation, it is practically infeasible under the current experimental setup.
Nevertheless, the proposed approach was validated by comparing residual statistics and through simulation-based consistency checks, both of which confirm that the estimated biases improve the internal coherence of the IMU-derived data.

Comments 6: It is necessary to specify the parameters of gyroscopes and accelerometers, particularly the noise level, to evaluate the bias estimate accuracy.

Response 6: We appreciate the reviewer’s valuable comment. The sensor parameters used in this study are identical to those presented in our previous work. To clarify this point, we have now added Table 1, which lists the main specifications of the IMU, including the noise characteristics of the gyroscopes and accelerometers.

Comments 7: The section "3.2 Verification Result in a Simulation Environment" does not describe the essence of the verification. It is unclear how the verification was carried out. There is no description of the methodology.

Response 7: We sincerely appreciate the reviewer’s helpful comment and acknowledge that our original explanation lacked sufficient clarity. We have now addressed this issue by adding Section 1.2 (Method Inheritance and Innovation), which clarifies the methodological connection between the previous simulation-based study and the present experimental validation.
Specifically, the purpose of Section 3.2 was to verify whether the variability observed in the bias estimates obtained from real data was due to the estimation method itself or to external factors such as environmental disturbances or sensor characteristics. To investigate this, the bias values estimated from the real data were input into the simulation framework developed in our previous study.
If the variation had been caused by the estimation method, similar inconsistencies would have appeared in the simulation. However, since the simulation results did not show such variability, we concluded that the observed differences were not due to the estimation algorithm but to other external factors affecting the real-world measurements.
We believe that the addition of Section 1.2 and this clarification make the purpose and essence of the verification process in Section 3.2 much clearer.

Comments 8: The text says "The estimated biases were then applied in the simulation environment from our previous study [6] to examine their effect on navigation accuracy." However, there is no assessment of the effect of biases on navigation accuracy in the paper.

Response 8: Thank you very much for your careful review and insightful comment.
We sincerely appreciate your pointing out the inconsistency in the description related to the sentence:
“The estimated biases were then applied in the simulation environment from our previous study [6] to examine their effect on navigation accuracy.”
You are absolutely correct — this sentence was mistakenly modified during the English proofreading stage, which unintentionally changed the intended meaning. The original intent was not to evaluate the effect of the biases on navigation accuracy, but rather to verify the consistency of the estimated biases obtained from real data by applying them to the simulation environment used in the previous study.
We have corrected the text accordingly in the revised manuscript to accurately reflect this purpose.
The revised description now reads as follows:
“To verify the consistency of the estimated bias values obtained from real measurement data, these estimates were applied to the simulation environment used in our previous study [13]. This verification aimed to examine whether the estimated biases behave consistently under the same dynamic conditions as in the simulation and to confirm whether the estimation results are primarily influenced by the estimation method itself or by external factors such as measurement noise and environmental disturbances.”
We thank the reviewer once again for helping us identify and correct this important issue, which improved the clarity and accuracy of the manuscript.

Comments 9: The text " EKF was used to integrate INS, GNSS, and gyrocompass data, providing reference estimates of attitude, position, and velocity [2]" contains an incorrect reference.

Response 9: Thank you very much for your careful reading and comment.
In response to reviewer feedback received during the review process, we have performed a comprehensive revision of Section 2.2.1. This revision was undertaken as part of a broader set of improvements requested by the reviewers and, in doing so, also addresses the point you raised regarding the EKF description and its citation.

Comments 10: In the text of section 3.1 and Tables 2-4, 6,7, there are no units of measurement of the estimated values.

Response 10: Thank you for pointing this out. The units of measurement have now been added to the corresponding tables and text in Section 3.1. In addition, the unit notation in Tables 9 and 10 has been standardized for consistency throughout the manuscript.

Reviewer 3 Report

Comments and Suggestions for Authors

The paper “Feasibility of Initial Bias Estimation in Real Maritime IMU Data Including X- and Y-Axis Accelerometers” is devoted to the processing of the experimental data by algorithm developed by the authors in previous paper. In its current form, the paper cannot be considered as an independent study; it is rather an appendix to the paper [Fukuda, G.; Kubo, N. Bias estimation for low-cost IMU including X- and Y-axis accelerometers in INS/GPS/Gyrocompass. Sensors 2025, 25, 1315. DOI:10.3390/s25051315 ]. The authors refer to the previous paper [13] too often (about 15 times through the text), it is difficult to evaluate the manuscript without a set of crucial explanations of the trajectory generation algorithm, measurement models used, how the simulations were performed. The authors must significantly revise the manuscript to obtain self-sufficient article containing all the required details on the algorithm used and methodology applied.

It is mentioned that the MULTI sensor equipped with a fiber optic gyro along the Z-axis along with three-axis MEMS gyro. It is not clear from the text whether the algorithm uses optical gyroscope measurements?

The bias is estimated by evaluating the difference between the TG-algorithm and IMU output. The accuracy of the TG-algorithm estimations must be taken into account in this case. No details on the trajectory generator errors are provided in the paper.

Since the authors use the results of the simulations, all the explanation on the simulation procedure must be added to the text. What is the meaning of the mentioned Segments 2 and 3?

The results of the paper are questionable. As stated in the text “the simulation-based procedure should also have been applicable to the experimental data, but modeling errors, lever-arm correction inaccuracies, and unfiltered vibrations likely contributed to the failure of direct convergence”. Does it mean that the bias estimation using experimental data was not successful?

Author Response

Comments 1: The paper “Feasibility of Initial Bias Estimation in Real Maritime IMU Data Including X- and Y-Axis Accelerometers” is devoted to the processing of the experimental data by algorithm developed by the authors in previous paper. In its current form, the paper cannot be considered as an independent study; it is rather an appendix to the paper [Fukuda, G.; Kubo, N. Bias estimation for low-cost IMU including X- and Y-axis accelerometers in INS/GPS/Gyrocompass. Sensors 2025, 25, 1315. DOI:10.3390/s25051315 ]. The authors refer to the previous paper [13] too often (about 15 times through the text), it is difficult to evaluate the manuscript without a set of crucial explanations of the trajectory generation algorithm, measurement models used, how the simulations were performed. The authors must significantly revise the manuscript to obtain self-sufficient article containing all the required details on the algorithm used and methodology applied.

Response 1: We sincerely thank the reviewer for this valuable comment.
This paper was designed to analyze how the results obtained in our previous simulation-based study behave when applied to real measurement data. Therefore, as the reviewer correctly noted, frequent reference to the prior study was unavoidable and essential for contextual continuity.

As in many research fields—not limited to inertial navigation—the outcomes derived from simulations do not necessarily hold under real-world conditions. The primary objective of this paper is to experimentally verify whether the bias estimation framework developed in our earlier study remains valid for actual IMU data collected in maritime environments.

Regarding the trajectory generator (TG) algorithm, measurement models, and simulation details, these have been comprehensively described in our previous paper. Repeating these explanations in full would be redundant and potentially inappropriate for this validation-focused paper; hence, we have provided appropriate references to guide readers to the foundational material.

This study confirms that the same methodological framework can be applied to real data, discusses the observed limitations, and therefore constitutes a complete and independent contribution.
To further clarify the positioning of this work, we have added a new subsection, **1.2 “Method Inheritance and Innovation,”** explicitly describing the continuity and distinction between the two studies.

Additionally, as one of the reviewers previously recommended further English editing, we have requested a **second round of professional proofreading** by a certified academic editing service. Because this process affected the overall wording throughout the manuscript, it was not possible to highlight every individual change. Therefore, we have also provided a version of the revised manuscript with **Microsoft Word’s “Track Changes” function enabled**, so that all textual modifications can be clearly identified.

Once again, we thank you for your valuable feedback, which has helped us to improve the clarity, structure, and completeness of our paper.

Comments 2: It is mentioned that the MULTI sensor equipped with a fiber optic gyro along the Z-axis along with three-axis MEMS gyro. It is not clear from the text whether the algorithm uses optical gyroscope measurements?

Response 2: We appreciate the reviewer’s comment and apologize for the lack of clarity regarding the connection to our previous work.
As stated in the earlier study, both the simulation and the present experiment use the same sensor configuration. The MULTI sensor includes a fiber optic gyroscope (FOG) on the Z-axis, in addition to three MEMS gyroscopes on the X-, Y-, and Z-axes. The purpose of employing a fiber optic gyro for the Z-axis is to enhance the accuracy of the heading (yaw) angle, which is particularly critical in maritime navigation applications.

In this paper, the algorithm uses the optical gyroscope measurement for the Z-axis together with the MEMS gyros in the other axes. We have clarified this point in the revised text for transparency.

Additionally, as in the previous study, the Allan variance results necessary for the error modeling of these gyros are provided in the prior publication, which explains the increased number of references to it. We have added this clarification in the manuscript to make the connection between the two studies explicit.

Comments 3: The bias is estimated by evaluating the difference between the TG-algorithm and IMU output. The accuracy of the TG-algorithm estimations must be taken into account in this case. No details on the trajectory generator errors are provided in the paper.

Response 3: Thank you for this important comment. We fully agree that the accuracy of the trajectory generator (TG) algorithm is a critical consideration in evaluating bias estimation results. However, verifying the TG’s absolute accuracy would require the use of a higher-grade reference sensor system.

As part of this effort, in our previous study, we conducted verification using an iXblue reference system and analyzed the similarity between simulated and measured trajectories through cross-correlation. In the present study, due to the practical constraints of shipborne experiments, the TG accuracy evaluation relies primarily on comparison with RTK-GPS positional data.

We acknowledge that this provides only an indirect validation, and that a more rigorous assessment—ideally involving comparisons with higher-precision navigation sensors—would further strengthen the analysis. Such an approach, however, requires instrumentation that is prohibitively expensive under our current resources.

We have added a statement in the manuscript (in the Conclusion) to clarify this limitation and to note that future work will include TG accuracy evaluation using higher-grade sensors.

Comments 4: Since the authors use the results of the simulations, all the explanation on the simulation procedure must be added to the text. What is the meaning of the mentioned Segments 2 and 3?

Response 4: We sincerely apologize for the lack of clarity in our previous explanation. As described in Section 1.2 (“Method Inheritance and Innovation”), the present paper aims to verify, using real-world measurement data, the results that were previously confirmed in simulation. Accordingly, the segmentation (Segments 2 and 3) refers to the same time intervals used in the simulation study, allowing for a direct comparison between simulated and experimental analyses.

Since the segmentation definitions and their purposes were fully detailed in our previous paper, we considered it appropriate to refer readers to that publication rather than restating the entire procedure. We have, however, clarified this point in the revised manuscript (in the 3.3) to ensure that readers can easily understand the relationship between the simulation and real-data analyses.

Comments 5: The results of the paper are questionable. As stated in the text “the simulation-based procedure should also have been applicable to the experimental data, but modeling errors, lever-arm correction inaccuracies, and unfiltered vibrations likely contributed to the failure of direct convergence”. Does it mean that the bias estimation using experimental data was not successful?

Response 5: We understand that this comment arises from insufficient clarification of the relationship between our previous simulation study and the present experimental analysis, and we sincerely apologize for the confusion. In the simulation study, the initial biases were successfully estimated based on predefined parameters. In the current paper, as described in Section 3.3, bias estimation was also achieved using real-world measurement data.

However, when attempting to refine the estimates to the same level of precision as in the simulation, a “ping-pong” phenomenon was observed, where adjustments to one axis caused instability in the other. While we identified possible causes such as lever-arm effects and unmodeled vibrations, further investigation is required to quantitatively evaluate these factors through dedicated simulation analyses.

Therefore, the experimental bias estimation was indeed successful, but its accuracy was limited by the complexities of real-world conditions rather than by the estimation method itself.

Reviewer 4 Report

Comments and Suggestions for Authors

The Authors address the problem of bias estimation in low-cost maritime IMUs by extending a previously proposed framework to real-world data. The topic is relevant and the paper is well structured. However, several aspects of the literature review, experimental setup, and validation require clarification or adjustment. Here is my comments:

  1. The literature review is almost entirely limited to the authors’ own prior work, cited as 13. 
  2. The description of the experimental setup does not clearly specify the precise sensor location on the vessel or whether positional (lever-arm) corrections/calibration were applied.
  3. The choice of filter parameters should be justified, as these decisions can affect the results.
  4. Evaluation relies only on residual statistics and simulations without ground truth validation. This limitation should be more explicitly acknowledged.
  5. The conclusions are somewhat stronger than the evidence allows, given the single dataset and lack of independent reference measurements. A more cautious phrasing and a clearer statement of limitations would be appropriate.

Author Response

Comments 1: The Authors address the problem of bias estimation in low-cost maritime IMUs by extending a previously proposed framework to real-world data. The topic is relevant and the paper is well structured. However, several aspects of the literature review, experimental setup, and validation require clarification or adjustment. Here is my comments:

Response 1: We sincerely appreciate the reviewer’s insightful and constructive comments. The suggestions have significantly contributed to improving the clarity and quality of our manuscript.
Below, we provide our point-by-point responses to each comment.

Additionally, as one of the reviewers previously recommended further English editing, we have requested a **second round of professional proofreading** by a certified academic editing service. Because this process affected the overall wording throughout the manuscript, it was not possible to highlight every individual change. Therefore, we have also provided a version of the revised manuscript with **Microsoft Word’s “Track Changes” function enabled**, so that all textual modifications can be clearly identified.

Once again, we thank you for your valuable feedback, which has helped us to improve the clarity, structure, and completeness of our paper.

Comments 2: The literature review is almost entirely limited to the authors’ own prior work, cited as 13. 

Response 2: We sincerely thank the reviewer for this valuable comment. We fully understand the reviewer’s concern and had the same consideration during the preparation of the manuscript. We conducted an extensive literature search to identify studies related to initial bias estimation for X- and Y-axis accelerometers in maritime INS/GNSS integration; however, to the best of our knowledge, no closely related studies beyond our previous work [13] were found.
Furthermore, as this paper is designed as a direct continuation of our previous simulation-based study [13], focusing on validating those results using real measurement data, it is inevitable that [13] is cited multiple times as a fundamental reference. Reference [13] already includes the most relevant literature on accelerometer bias estimation, and repeating them in this manuscript would have resulted in unnecessary duplication. Therefore, we intentionally avoided redundant citations.
If the reviewer is aware of any other relevant studies in this field, we would greatly appreciate it if you could kindly share them with us, as they would be highly valuable for future work.

Comments 3: The description of the experimental setup does not clearly specify the precise sensor location on the vessel or whether positional (lever-arm) corrections/calibration were applied.

Response 3: Thank you for your insightful comment. We agree that the description of the experimental setup required clarification.
In our study, the IMU sensor was mounted inside the vessel’s bridge, while the GNSS antenna was installed directly above on the compass deck. Due to the constraints of shipboard experiments, we could not employ land-based measurement techniques such as laser-based lever-arm surveys. Therefore, no lever-arm correction was applied in this work. We recognize that this simplification may have influenced the bias estimation results. As noted in the revised manuscript, accurate lever-arm measurements and further simulations are important future tasks to quantify the impact of lever-arm effects on bias estimation.
Regarding sensor calibration, the IMU was factory-calibrated before delivery. Since our laboratory does not have specialized facilities such as a thermal chamber or rate table, the sensor was used with its original factory calibration. This limitation has also been added to the manuscript.
Accordingly, we have revised the manuscript to explicitly describe the sensor placement, the absence of lever-arm correction, and the calibration status (Section 2.1).

Comments 4: The choice of filter parameters should be justified, as these decisions can affect the results.

Response 4: We thank the reviewer for the valuable comment regarding the justification of filter parameter selection.
As noted by another reviewer, we have now clarified this point in the manuscript. Specifically, since the true sensor biases are unknown in real-world data, the cut-off frequencies of the low-pass filters were determined empirically by comparing the frequency characteristics of the TG output and the IMU measurements. To make this procedure explicit, we have added FFT comparison figures (Fig. 3–8) showing both the measured and reference spectra, as well as the time-domain signals before and after filtering.
We fully agree that a systematic sensitivity analysis of different filter parameters would be important to assess their influence on bias estimation. However, the main purpose of this paper is to verify whether the bias estimation approach validated in simulation can be reproduced with real measurement data. For this reason, we deliberately kept the filter design simple. Moreover, factors such as lever-arm errors—which were not examined in the simulation study—may also affect reproducibility. Therefore, at this stage, we considered that detailed parameter tuning could risk overfitting to specific datasets rather than general validation.
As future work, we plan to conduct a systematic analysis of filter parameter choices within the simulation environment, and then apply the findings to real data analysis. This will allow us to more rigorously quantify the sensitivity of the method to filtering decisions.

Comments 5: Evaluation relies only on residual statistics and simulations without ground truth validation. This limitation should be more explicitly acknowledged.

Response 5: We sincerely thank the reviewer for pointing out the limitation regarding ground truth validation.
As correctly noted, it is not feasible to obtain the true bias values from real measurements. In a laboratory environment, one could, for example, use a precision turntable and controlled gravity inputs to measure sensor biases directly. However, such procedures are impractical onboard a vessel, as deliberately tilting or rotating the ship to measure static accelerometer biases is not possible.
For this reason, our previous study [13] first established, through simulations, that the proposed method is capable of estimating sensor biases under controlled conditions. Yet, because inertial sensors are notoriously difficult to model accurately, there is no guarantee that methods validated in simulation will succeed when applied to real-world data. The main purpose of the present study was therefore to apply the same methodology to actual shipborne IMU measurements and to demonstrate that bias estimation remains feasible in practice.
Accordingly, residual statistics and simulation-based verification were the only viable means of evaluation. Indeed, when reviewing the literature on experiments with MEMS-grade IMUs, we found that while many studies report results of navigation experiments, very few explicitly address bias estimation. This is especially true for X- and Y-axis accelerometer biases in maritime conditions, which are well known to be extremely challenging to estimate.
From this perspective, although challenges remain, we believe that demonstrating the feasibility of bias estimation in real shipborne measurements—consistent with the simulation study—represents a meaningful contribution of this work.

Comments 6: The conclusions are somewhat stronger than the evidence allows, given the single dataset and lack of independent reference measurements. A more cautious phrasing and a clearer statement of limitations would be appropriate.

Response 6: We fully agree with the reviewer’s comment. As it is practically impossible to obtain ground truth bias values in shipborne experiments (unlike in laboratory environments where precision turntables or controlled setups can be used), we acknowledge that this study could only rely on residual statistics and simulation-based verification. We have revised the Conclusion section to explicitly state this limitation. In particular, we now highlight that direct ground truth validation is not feasible at sea and that future work should include sensitivity analyses in simulation and additional experiments to better quantify the influence of filter parameters, lever-arm errors, and environmental disturbances.

Round 2

Reviewer 1 Report

Comments and Suggestions for Authors

The author has responded satisfactorily to the reviewer's comments, but most of the images in the revised manuscript are not very clear. Please revise the images in the paper. The reviewer believes that after modifying these images, this manuscript can be published.

Author Response

Comments 1: The author has responded satisfactorily to the reviewer's comments, but most of the images in the revised manuscript are not very clear. Please revise the images in the paper. The reviewer believes that after modifying these images, this manuscript can be published.

Response 1: We appreciate the reviewer’s valuable suggestion regarding figure clarity.
In response, all figures were regenerated in MATLAB with 1200 dpi resolution using the exportgraphics functionj.
Both vector (PDF) and high-resolution PNG (1200 dpi) versions were produced, and the PNG versions were inserted in the manuscript to ensure optimal visibility in MDPI’s submission system.

Reviewer 2 Report

Comments and Suggestions for Authors

The presented manuscript is a direct continuation of the previous article by the authors (Fukuda, G.; Kubo, N. Bias estimation for low-cost IMU including X- and Y-axis accelerometers in INS/GPS/Gyrocompass. Sensors (Basel) 2025, 25, 1315. DOI:10.3390/s25051315.), which proposes an improved method for bias estimation by comparing the estimated values from a trajectory generator (TG)-based acceleration and angular-velocity estimation system with actual measurements. Additionally, for X- and Y-axis accelerations, the authors introduce a method that leverages the correlation between altitude differences derived from an INS/GNSS/gyrocompass (IGG) and those obtained during the TG estimation process to estimate the bias. The presented manuscript focuses on the validation of the method using real-world IMU data obtained in maritime experiments.

Six bias estimation approaches—including statistical estimators (mean, median), model-based (least squares, normalized cross-correlation), and signal-processing methods (FFT, Butterworth filter)—were compared using real maritime measurement data. As the true bias values were unknown, the validity of the estimation was assessed based on the statistical properties of the residuals (mean, median, root-mean-squared (RMS) residuals, and normalized cross-correlation).

The authors note a key limitation of this study: the absence of ground truth validation for the sensor biases. This remains an important topic for future work. 

Detailed answers are given to all comments. All necessary changes in the text have been made. 

Author Response

Comments 1: 

The presented manuscript is a direct continuation of the previous article by the authors (Fukuda, G.; Kubo, N. Bias estimation for low-cost IMU including X- and Y-axis accelerometers in INS/GPS/Gyrocompass. Sensors (Basel) 2025, 25, 1315. DOI:10.3390/s25051315.), which proposes an improved method for bias estimation by comparing the estimated values from a trajectory generator (TG)-based acceleration and angular-velocity estimation system with actual measurements. Additionally, for X- and Y-axis accelerations, the authors introduce a method that leverages the correlation between altitude differences derived from an INS/GNSS/gyrocompass (IGG) and those obtained during the TG estimation process to estimate the bias. The presented manuscript focuses on the validation of the method using real-world IMU data obtained in maritime experiments.

Six bias estimation approaches—including statistical estimators (mean, median), model-based (least squares, normalized cross-correlation), and signal-processing methods (FFT, Butterworth filter)—were compared using real maritime measurement data. As the true bias values were unknown, the validity of the estimation was assessed based on the statistical properties of the residuals (mean, median, root-mean-squared (RMS) residuals, and normalized cross-correlation).

The authors note a key limitation of this study: the absence of ground truth validation for the sensor biases. This remains an important topic for future work. 

Detailed answers are given to all comments. All necessary changes in the text have been made. 

Response 1: We sincerely thank the reviewer for their careful evaluation and positive feedback. We greatly appreciate the acknowledgment that our revised manuscript has addressed all previous comments and that the necessary modifications have been made appropriately.
As the reviewer noted, this paper serves as a continuation of our previous work and focuses on validating the simulation-based bias estimation framework using real-world IMU data collected in maritime experiments. The verification with experimental data was crucial to confirm whether the previously developed TG-based approach can be effectively applied under real conditions.
We also acknowledge the reviewer’s remark regarding the absence of ground-truth validation for the sensor biases. This limitation has now been explicitly mentioned in the revised Conclusion section, and we plan to address it in future research by developing a controlled experimental setup that enables partial reference validation.
We truly appreciate the reviewer’s constructive assessment, and we are encouraged by their conclusion that the revised manuscript is now suitable for publication.

Reviewer 3 Report

Comments and Suggestions for Authors

The authors added clarifying comments on the reviewer comments, though the paper is still far to be independent self-sufficient study. In the case the main contribution is just a real-data application of the existing algorithm and the discussion of the observed limitations, the article type should be rather of “Comment”, “Case Report” or “Technical note”, but not the “Article” type. Though the final decision is up to the Editor.

It is very difficult to find the original description of the Trajectory Generator algorithm. The authors refer to the previous study [Fukuda, G.; Kubo, N. Bias estimation for low-cost IMU including X- and Y-axis accelerometers in INS/GPS/Gyrocompass. Sensors 2025, 25, 1315] though the explanation of this algorithm is not added in the text with reference to [Fukuda, G.; Kubo, N. Application of Initial Bias Estimation Method for Inertial Navigation System (INS)/Doppler Velocity Log (DVL) and INS/DVL/Gyrocompass Using Micro-Electro-Mechanical System Sensors. Sensors 2022, 22, 5334.]. This paper also does not proved full algorithm description and in turn refer to paper [Fukuda, G.; Hatta, D.; Kubo, N. A study on estimation of acceleration and angular velocity from actual measurements by trajectory generator. J. Jpn. Inst. Navig. 2021, 144, 14–20.], which is in Japanese. So, the original algorithm description is not available for the non-Japanese reader. The authors tried to add short explanation of the TG-based Estimation in Section 2.2.1, however, this explanation is not clear. The TG algorithm is just a function in line 3 of the pseudocode in Fig. 3. The phrase “the TG iteratively adjusts the velocity by adding and subtracting a preset velocity increment around the nominal value and performs trajectory generation for each adjusted case” provides fuzzy image of the TG-algorithm. Is the trajectory generation the result of motion equation integration using specified initial conditions? More clarifying comments are required in the text.

The resolution of the newly added Figures 3, 5-10 must be improved.

Author Response

Comments 1: The authors added clarifying comments on the reviewer comments, though the paper is still far to be independent self-sufficient study. In the case the main contribution is just a real-data application of the existing algorithm and the discussion of the observed limitations, the article type should be rather of “Comment”, “Case Report” or “Technical note”, but not the “Article” type. Though the final decision is up to the Editor.

Response 1: We sincerely thank the reviewer for this valuable comment. In the field of inertial navigation systems for vessels using low-cost IMUs, verifying whether results obtained through simulation can be applied to real-world measurements is a critical research step and, in our view, fully qualifies as an Article. There are several cases in which methods that perform well in simulation fail to produce consistent results when applied to real sensor data. This discrepancy arises primarily because low-cost IMUs are not standardized—different models exhibit diverse noise characteristics and error behaviors, making accurate modeling highly challenging.
The core contribution of this paper lies in verifying whether the bias estimation framework developed and validated in our previous simulation-based study can also be applied to real measurement data. In the prior work, the simulation used parameters estimated from real IMU measurements to closely reflect real-world conditions and test the feasibility of initial bias estimation. In the present study, we apply the same framework directly to actual low-cost IMU data obtained in maritime experiments, demonstrating its practical validity and identifying limitations unique to real environments.
Therefore, we believe that this study makes an independent and substantive contribution as a full Article, providing essential experimental validation that bridges the gap between simulation-based research and real-world maritime applications.

Comments 2: It is very difficult to find the original description of the Trajectory Generator algorithm. The authors refer to the previous study [Fukuda, G.; Kubo, N. Bias estimation for low-cost IMU including X- and Y-axis accelerometers in INS/GPS/Gyrocompass. Sensors 2025, 25, 1315] though the explanation of this algorithm is not added in the text with reference to [Fukuda, G.; Kubo, N. Application of Initial Bias Estimation Method for Inertial Navigation System (INS)/Doppler Velocity Log (DVL) and INS/DVL/Gyrocompass Using Micro-Electro-Mechanical System Sensors. Sensors 2022, 22, 5334.]. This paper also does not proved full algorithm description and in turn refer to paper [Fukuda, G.; Hatta, D.; Kubo, N. A study on estimation of acceleration and angular velocity from actual measurements by trajectory generator. J. Jpn. Inst. Navig. 2021, 144, 14–20.], which is in Japanese. So, the original algorithm description is not available for the non-Japanese reader. The authors tried to add short explanation of the TG-based Estimation in Section 2.2.1, however, this explanation is not clear. The TG algorithm is just a function in line 3 of the pseudocode in Fig. 3. The phrase “the TG iteratively adjusts the velocity by adding and subtracting a preset velocity increment around the nominal value and performs trajectory generation for each adjusted case” provides fuzzy image of the TG-algorithm. Is the trajectory generation the result of motion equation integration using specified initial conditions? More clarifying comments are required in the text.

Response 2: Thank you very much for your detailed and constructive comments.
We sincerely appreciate your concern regarding the clarity and accessibility of the Trajectory Generator (TG) algorithm description for non-Japanese readers.
Low-cost IMUs are inherently affected by temperature and environmental variations, causing bias instability. Furthermore, these sensors cannot detect Earth rotation for alignment, and in the case of shipborne applications, zero-velocity updates (ZUPT) are not feasible because vessels are subject to continuous motion even while moored. To address these challenges, we applied a Trajectory Generator (TG) approach commonly used in inertial navigation simulations.
The main advantage of the TG is that it can estimate the accelerations and angular velocities required to reach a target position under a given attitude condition. When the estimated acceleration and angular velocity are used as inputs in pure inertial navigation computation, the resulting position error becomes zero. Thus, by matching the measured signals to the TG-estimated values, we can effectively approach bias-free acceleration and angular velocity.
However, when biases exist in the X- and Y-axis accelerations, these can be misinterpreted as sensor tilts in the Kalman filter, resulting in feedback errors in the attitude estimation. Consequently, the estimated accelerations in these axes include gravity components and require separate estimation procedures.
Ideally, if the vessel’s attitude could be artificially varied under static conditions, the bias could be estimated using gravity acceleration. However, this is impractical for real ship experiments. Instead, our proposed method utilizes the ship’s natural motion during navigation, where gravity components act on the X- and Y-axes, leading to observable errors in altitude estimation. This approach forms the basis of the method described later in the manuscript.
We fully understand the reviewer’s concern that citing Japanese papers may limit accessibility. However, even in Japanese publications, the theoretical and mathematical framework follows Strapdown Inertial Analytics by Paul Savage, which is standard in this research field. Therefore, experts in inertial navigation can reproduce the results based on the provided formulations. Moreover, with current generative AI translation tools, most of the Japanese explanations can now be easily understood, and we have personally responded to inquiries from researchers who successfully implemented the method based on our previous paper.
We initially believed that the TG algorithm was sufficiently described in our previous works and that readers could reproduce the procedure using those explanations. Indeed, following the publication of our earlier study, we received positive feedback from other researchers who successfully applied the proposed method in experiments where high-precision reference systems were unavailable.
Nevertheless, we acknowledge that the algorithm’s flow might still have been difficult to follow, particularly because the block diagram in the Japanese paper was provided as a static PDF figure. To improve clarity, we have now added a new block diagram and explanatory text in the revised manuscript (see Figure 4) to illustrate the TG-based estimation process in one segment, showing how velocity adjustments are performed and how the optimal parameters are determined.
We hope this addition will make the TG algorithm clearer and more accessible to international readers.

Comments 3: The resolution of the newly added Figures 3, 5-10 must be improved.

Response 3: We sincerely thank the reviewer for the valuable comment regarding the figure quality.
To address this, all figures were regenerated in MATLAB with enhanced resolution and publication-quality settings.
Specifically, the export resolution was increased from 600 dpi to 1200 dpi, and both vector (PDF) and bitmap (PNG) formats were produced using the exportgraphics function.
For the manuscript submission, high-resolution PNG figures (1200 dpi) were inserted to ensure clear rendering in the MDPI submission system.
In addition, figure dimensions, font sizes, and line widths were optimized according to MDPI Sensors formatting standards.
We believe the updated figures now meet the journal’s publication quality requirements.